# Nanobubble-actuated ultrasound neuromodulation for selectively shaping behavior in mice

Xuandi Hou [1,3], Jianing Jing[1,3], Yizhou Jiang[1], Xiaohui Huang[1], Quanxiang Xian[1], Ting Lei[1], Jiejun Zhu[1,2], Kin Fung Wong[1], Xinyi Zhao[1], Min Su[1], Danni Li[1], Langzhou Liu [1], Zhihai Qiu[2] & Lei Sun [1] ✉

Ultrasound is an acoustic wave which can noninvasively penetrate the skull to deep brain regions, enabling neuromodulation. However, conventional ultrasound's spatial resolution is diffraction-limited and low-precision. Here, we report acoustic nanobubble-mediated ultrasound stimulation capable of localizing ultrasound's effects to only the desired brain region in male mice. By varying the delivery site of nanobubbles, ultrasound could activate specific regions of the mouse motor cortex, evoking EMG signaling and limb movement, and could also, separately, activate one of two nearby deep brain regions to elicit distinct behaviors (freezing or rotation). Sonicated neurons displayed reversible, low-latency calcium responses and increased c-Fos expression in the sub-millimeter-scale region with nanobubbles present. Ultrasound stimulation of the relevant region also modified depression-like behavior in a mouse model. We also provide evidence of a role for mechanosensitive ion channels. Altogether, our treatment scheme allows spatially-targetable, repeatable and temporally-precise activation of deep brain circuits for neuromodulation without needing genetic modification.

In recent years, there has been a suite of different neuromodulation techniques developed for probing neural systems and treating brain diseases. A rapidly-growing modality, ultrasound (US), has been harnessed for a myriad of neuromodulation applications. Ultrasound is a wave of mechanical energy, able to non-invasively target deep-seated regions in the brain and to be focused down to spots a few millimeters wide. US is, moreover, compatible with MRI for safe, simultaneous monitoring of modulated brain activity[1–3]. Ultrasound has been used to manipulate the activity in brain regions of several animal species, including rodents[4], rabbits[5], sheep[6], and non-human primates[7]. US has also been studied as a potential tool to treat a variety of neurological diseases in humans, such as Alzheimer's disease[8], epilepsy[9], Parkinson's disease[10,11], and depression[12]. Ultrasound, therefore, is a

promising, minimally-invasive means of manipulating central nervous system (CNS) function.

For all ultrasound's attributes as a neuromodulation modality, there are also some inherent problems with its application. The frequencies of ultrasound that would be needed to safely penetrate an intact skull are relatively low; consequently, the matching diffraction-caused spatial precision would lie between a millimeter and a centimeter[13]. The acoustic heterogeneity and erratic physical features of diverse skulls can also affect ultrasound focusing[14]. These elements are hurdles in the path of achieving a properly-focused ultrasonic beam which activates only the desired neural circuit or brain region. Another complication is that the brain has some inherent sensitivity to ultrasound that is unevenly distributed in different areas, via

[1]Department of Biomedical Engineering, The Hong Kong Polytechnic University, Hung Hom 999077 Hong Kong SAR, PR China. [2]Guangdong Institute of Intelligence Science and Technology, Hengqin, Zhuhai 519031 Guangdong, China. [3]These authors contributed equally: Xuandi Hou, Jianing Jing. ✉e-mail: lei.sun@polyu.edu.hk

mechanisms such as mechanosensitive ion channels[15,16], cytoskeleton, and cell-extracellular matrix communications[17]. As a result, some brain regions with low mechanosensitivity may be challenging to successfully probe with an ultrasonic stimulus. Thus, it may not always be possible to achieve well-targeted neuromodulation of a particular brain or brain region with US-alone.

Several workarounds to these issues have been developed, such as sonogenetics, laser-induced ultrasound, and nanoparticle-based ultrasound stimulation approaches[18–20]. These emerging methods are a valuable improvement as they can significantly improve the accuracy and spatial resolution of non-invasively-delivered ultrasound. Although sonogenetics is a promising candidate for brain stimulation, it relies on virally-transfecting CNS neurons, which has so far limited its applications in human brains[21–24]. Laser-induced ultrasound stimulation provides good spatial resolution, but still requires optical energy to be delivered locally - i.e., invasively - to the targeted site through fibers for the duration of stimulation[20]. Nanoparticles are another approach to US neuromodulation, with the advantage that they do not require genetic modification. Nanoparticle-mediated techniques generally rely on nanomaterials to convert a remotely-transmitted primary stimulus (light, US, magnetism, etc) to a localized secondary stimulus at the nanomaterial-neuron interface. The secondary signal is only transmitted where nanoparticles are present, thus allowing more targeted neuromodulation. Therefore, applying lessons from nanotechnology in this context could enable a new generation of minimally-invasive and highly-targeted neuromodulation approaches to be developed[25]. Among the methods employing nanomaterials as mediators to improve the specificity or reduce the degree of invasion[26–29], nanobubble-mediated ultrasound neuromodulation could directly localize and amplify the ultrasonic effects for spatially accurate neuronal modulation of a targeted deep brain region[18]. Since the localized neuromodulatory effect depends primarily upon the presence of nanobubbles, the limited focusing ability of ultrasound may be circumvented without needing image guidance. Given that the US effect can be localized and amplified by nanobubbles, we wished to explore whether this all-acoustic US+nanobubbles method could precisely

modulate specific neural circuits, induce distinct animal behaviors, and rescue neurological conditions.

In this paper, we present nanobubbles with optimized surface properties (PEGylated gas vesicles, PGVs), as ultrasonic actuators for localized neural stimulation in an acoustic field. We show that the PGVs+US strategy evokes reversible Ca²⁺ signaling through mechanosensitive ion channels, and the location of neuronal activation was largely co-located with PGVs in targeted brain regions, which showed significantly greater c-Fos expression than their respective contralateral regions, the auditory cortex, as well as other brain areas without PGVs. We also show robust and repeatable ultrasonic activation of limb movement in anesthetized mice with PGVs-injected cortices. Ultrasound stimulation of deeper brain regions with PGVs allowed could activate specific neural circuits in awake, freely-moving mice to evoke defined behaviors. We also demonstrated some therapeutic effects of PGVs+US stimulation by selectively activating 5-HT neurons in the dorsal raphe nucleus (DRN) and reducing depression-like behaviors in mice. No cell damage and inflammation were observed in vitro or in vivo after stimulation, and PGVs have negligible effects on the mouse brain's normal function. Altogether, we demonstrate a remote, nanobubble-enabled, non-genetic toolkit for selective neuromodulation with low-frequency low-intensity ultrasound, with multiple possible applications.

## Results

### Characterization of PGVs' properties

Nano gas vesicles were produced from *Anabaena flos-aquae* using hypertonic lysis[30]. The surface of GVs is eminently suitable for chemical modification, which makes it possible to modify their properties through relatively simple means. To improve water solubility and biocompatibility, the polymer was coated with an amphiphilic, FDA-approved polymer shell, poly(ethylene glycol) (PEG). PGVs have an average width between 50–100 nm, and lengths ranging from 200 nm to 600 nm, as revealed by transmission electron microscopy (TEM, Fig. 1a). The PEG coating was not visible in TEM imaging due to the low contrast between protein shell and PEG[31]. In contrast, the slightly

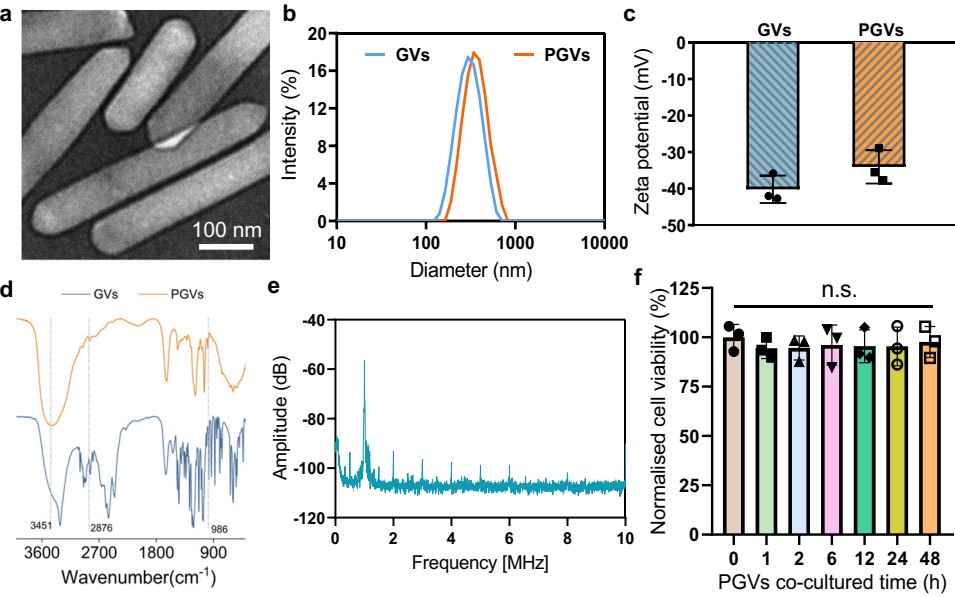

**Fig. 1 | Basic characterization of the prepared PGVs. a** Transmission electron micrographs of PGVs. The representative image was from *n* = 3 independent experiments. **b** Size distribution plot for GVs and PGVs observed by dynamic light scattering. **c** Zeta potential of GVs and PGVs in DI H₂O. Bars represent the mean ± SD of 3 independent experiments. **d** FTIR spectra of GVs, and PGVs. **e** Averaged frequency spectrum of backscattered signals from PGVs suspension with 3 independent tests. **f** Cytotoxicity of PGVs (0.8 nM), measured by an MTT test. Primary neurons were exposed to PGVs in a culture medium for the stated amount of time. Bars represent the mean ± SD of 3 independent experiments. $p > 0.05$ ($P_{1h/0h} = 0.8860$, $P_{2h/0h} = 0.9007$, $P_{6h/0h} = 0.9743$, $P_{12h/0h} = 0.9500$, $P_{24h/0h} = 0.9465$, $P_{48h/0h} = 0.9977$), no significant differences (n.s.), in one-way ANOVA. Source data are provided as a Source Data file.

altered (~46 nm) diameter of PGVs was detectable by dynamic light scattering (DLS, Fig. 1b). The zeta potential of PGVs was −34.1 ± 4.5 mV, higher than GVs (−40.2 ± 3.7 mV), suggesting successfully-conjugated PEG on the GVs' surface (Fig. 1c). FTIR spectroscopy was performed to confirm the presence of functional groups PEG over the surface of GVs. The characteristic peaks of PGVs could be ascribed to 3451 cm$^{-1}$ for O-H stretching vibrations, indicating the successful PEGylation (Fig. 1d). The additional infrared peaks 986 cm$^{-1}$ and 2876 cm$^{-1}$ can be considered as PEG in-plane deformation vibration peaks of C-O-C, and the symmetrical stretching vibration peaks of -CH$_2$, respectively. No corresponding peaks could be seen in the GVs-alone group. Thus, the data indicated that the GVs were successfully loaded with PEG, making PGVs.

The local oscillation in the PGVs' vicinity was measured by a hydrophone. We carefully analyzed the experimental acoustic and cavitation of these PGVs and observed the backscattered signals in the time- and frequency-domains to monitor produced patterns of cavitation. No broadband signal was seen, and only the 1st to 8th harmonic signals appeared (Fig. 1e), indicating that no inertial cavitation occurred when the PGVs were sonicated in our setup. We also found that our prepared PGVs were not cytotoxic on their own to primary neurons in culture (Fig. 1f).

## Ultrasound mechanically activates neurons through stable PGVs

GVs are a novel kind of biogenic nanobubble, which are highly stable and can serve as genetically-encoded ultrasound contrast agents to track targeted microbes or mammalian cells by ultrasound imaging[18,32,33]. Based on our group's previous experiences using GVs for neuromodulation, we were particularly interested in further improving the stability of GVs in vivo, in order to enable the repetition of ultrasound stimulation over a longer period of time. To identify the stability of GVs after surface-coated PEG in vivo, we employed two types of imaging modalities for the nanobubbles' lifetime test - ultrasound imaging and IVIS imaging (Fig. 2a). Both GVs- and PGVs-injected areas produced robust ultrasound contrast signals relative to other tissue (Fig. 2b, Supplementary Fig. 1). Ultrasound contrast signals in GVs and PGVs decreased as the post-injected time increased, indicating their biodegradability. In addition, PGVs showed echogenicity at 12-day post-injection, which was much longer than GVs-alone (6 days) indicating that PEG surface modification improved the stability of GVs in vivo (Fig. 2c). PGVs stability was further confirmed by IVIS imaging with ICG-labeled PGVs/GVs (Fig. 2c, d, Supplementary Fig. 2a–d). Successful conjugation of ICG-PGVs and ICG-GVs was validated via absorbance peak and solution color change (from milky white to cyan) (Supplementary Fig. 2b, c). Compared to free ICG, the absorption spectra of ICG-linked PGVs/GVs were significantly broadened, indicating their successful binding (Supplementary Fig. 2c). This stability of PGVs indicated that they were likely suitable for the in vivo studies that follow this section, since mice injected with GVs or PGVs need a few days of post-operative recovery before experiments.

To investigate how neurons respond to ultrasound in acoustic conditions that resemble soft tissue, we conducted experiments using primary cortical neurons cultured on an acoustically transparent polyacrylamide (PA) gel. Calcium responses of these cultured neurons to ultrasound were optically recorded (Supplementary Fig. 3a). To minimize the formation of standing waves, we positioned the cultured neurons at the base of a water tube and submerged an ultrasound transducer in degassed water. We utilized a 1.0 MHz transducer, a frequency within the range employed in recent studies involving various organisms, ensuring uniform delivery of ultrasound to neurons within our field of view, and pulsed-wave ultrasound stimulation to minimize the possible thermal effects (Supplementary Fig. 4a). Next, we evaluated whether PGVs could mediate ultrasound stimulation to achieve neuromodulation. Ultrasound is a form of mechanical energy, capable of acting on mechanosensitive ion channels to create an ion influx. The effect of GVs/PGVs is also mechanical in nature, locally amplifying the mechanical energy transmitted by ultrasound. Therefore, we hypothesized that the effect of PGVs+US on neurons was through the influx of cations through mechanosensitive ion channels (Supplementary Fig. 3a). Dynamic calcium imaging of primary neurons during PGVs+US stimulation was performed, with a fixed inter-pulse interval of 10 s to allow Ca$^{2+}$ levels to reduce between stimuli. Reversible and repeatable Ca$^{2+}$ influx with high amplitude was detected in the PGVs+US-sonicated neurons (Supplementary Fig. 3b, c, Supplementary Movie 1). The maximum increase in fluorescence intensity ($\Delta F/F_0$) for PGVs+US was 46.2% on average, whereas no obvious changes with US-alone, or in either condition before US pulses were delivered (Supplementary Fig. 3c).

Having established that ultrasound's effects were exerted through cellular calcium influx, the role of mechanosensitive ion channels was then studied. To block mechanosensitive ion channels, we selected gadolinium (Gd$^{3+}$) as a general blocker. Gd$^{3+}$ is known to alter the deformability of the lipid bilayer, thereby inducing changes in membrane mechanics and effectively inhibiting mechanosensitive ion channels[34]. To ensure the specificity of our study, we utilized a concentration of 20 μM Gd$^{3+}$ to avoid blocking non-mechanosensitive channels or causing any significant changes to cell excitability[35]. In the presence of Gd$^{3+}$, we observed a substantial decrease in the amplitude of the evoked responses, reducing to 33.8%, and recovered to 87.9% upon Gd$^{3+}$ washout (Supplementary Fig. 3d, e). Thus, we confirmed that PGVs+US acts on neurons in significant part through the activation of mechanosensitive ion channels. The partial persistence of the Ca$^{2+}$ response could potentially be attributed to calcium release from internal sources (e.g. ER, mitochondria) triggered by PGVs+US[18]. Finally, we conducted investigations to determine whether the neuronal response to PGVs+US was intrinsic to the neurons themselves (cell-autonomous) or if it relied on synaptic connections with astrocytes or excitatory neurons. Neurons were pre-treated with the post-synaptic blockers NBQX (10 μM) and gabazine (20 μM) and then stimulated with PGVs+US. Our findings indicated that the application of signaling blockers did not have a significant impact on the neuronal response to ultrasound (93.7% of response in normal medium) (Supplementary Fig. 3f, g), indicating that the excitation induced by ultrasound stimulation did not heavily rely on synaptic transmission.

## PGVs-actuated US can trigger motor responses through targeted activation of the motor cortex

We next studied the ability of PGVs to be employed as acoustic mediators for spatially targeted stimulation of the motor cortex to manipulate mouse behaviors. To this end, PGVs were delivered by an injection into the right motor cortex (Fig. 3a). As the skull attenuates acoustic energy that passes through it, several US intensities between 0.17–0.54 MPa were tested, with a 40% duty cycle. Ultrasound evoked visible movements in the left forelimbs of PGVs-injected (referred to as 'PGVs$^+$') mice in an ultrasound intensity-dependent manner (Fig. 3b, c, Supplementary Movie 2). In contrast, no visible left forelimb movement was evoked in mice injected with saline (referred to as 'Saline$^+$') or collapsed PGVs (Fig. 3b, c). The response latency of forelimbs movement was approximately 249.7 ms (0.40 MPa), 160.9 ms (0.47 MPa), and 140.8 ms (0.54 MPa), indicating PGVs were better able to actuate neural stimulation under increased US intensity (Fig. 3d). Comparing left and right forelimb movement of the same PGVs$^+$ mouse, there was significantly greater movement seen in the left than in the right (Fig. 3e, Supplementary Movie 2). Further, evoked left forelimb movement showed dependence on US acoustic intensity, whereas the right showed almost no movement even at the highest US intensity (0.54 MPa) (Fig. 3f).

The electromyography (EMG) responses in the bilateral biceps brachii and biceps femoris muscles during US brain stimulation were also monitored. A 1.0 MHz ultrasound transducer was positioned

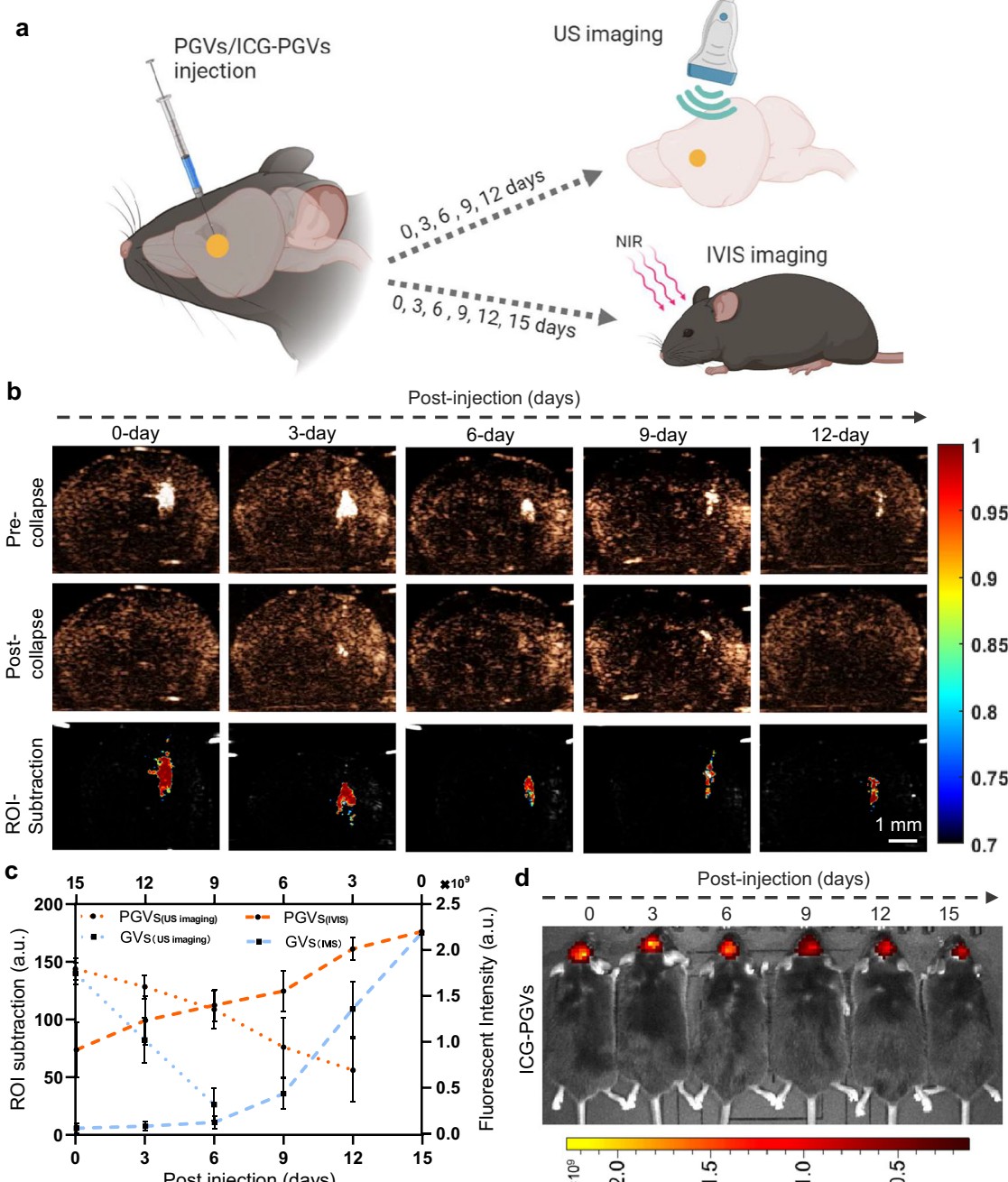

**Fig. 2 | Improved stability of PGVs in mice brains. a** Schematic illustration showing the basic experimental process. PGVs or ICG-PGVs were injected into the mice's brains imaging with different imaging modalities. **b** Ultrasound contrast-mode images of PGVs-injected mice brains at several injected days. PGVs collapsed with destructive insonation. **c** Quantitative results of the PGVs and GVs signals in mice brains by ultrasound imaging and IVIS. *a.u., arbitrary units*. Data represent the mean ± SEM from 3 independent experiments. **d** NIR fluorescent imaging of mice brains with several injected-time points (0, 3, 6, 9, 12, and 15 days) with brain ICG-PGVs injection. Source data are provided as a Source Data file.

above the head of an anesthetized mouse and an EMG probe connected to the triceps muscles in the left forelimb (Fig. 3g). We found low-intensity ultrasound stimulation evoked contractions in PGVs[+] mice, but mice with Saline or collapsed PGVs showed little to no response (Fig. 3h, i). In PGVs[+] mice, both the relative amplitude and rate of response were higher, and an overall pattern of ultrasonic intensity-dependent EMG responses was seen (Fig. 3i). These responses were of greater magnitude than those seen in our group's previous MscL-sonogenetics-induced EMG responses, suggesting that PGVs maybe even better candidate for assisting ultrasound brain stimulation[19]. Saline and collapsed PGVs mice did not exhibit any EMG

response under lower ultrasonic pressures of 0.05 and 0.09 MPa but did exhibit modest responses at higher pressures (0.17 MPa) (Fig. 3h, i). In PGVs[+] mice, the latency of EMG responses (<160 ms) decreased with the increased applied energy of ultrasound, but without significant differences (Fig. 3j).

The capability of the PGVs-actuated US setup to activate neurons was also evaluated by examining the expression of the activation marker c-Fos[36,37]. Confocal imaging confirmed the presence of PGVs (DiO[+]-PGVs, green) in the targeted cortex region (Fig. 3k). Only the area of the brain where PGVs were present showed elevated c-Fos levels, but not others (Fig. 3k, l Supplementary Fig. 5a–c). The number of c-Fos-

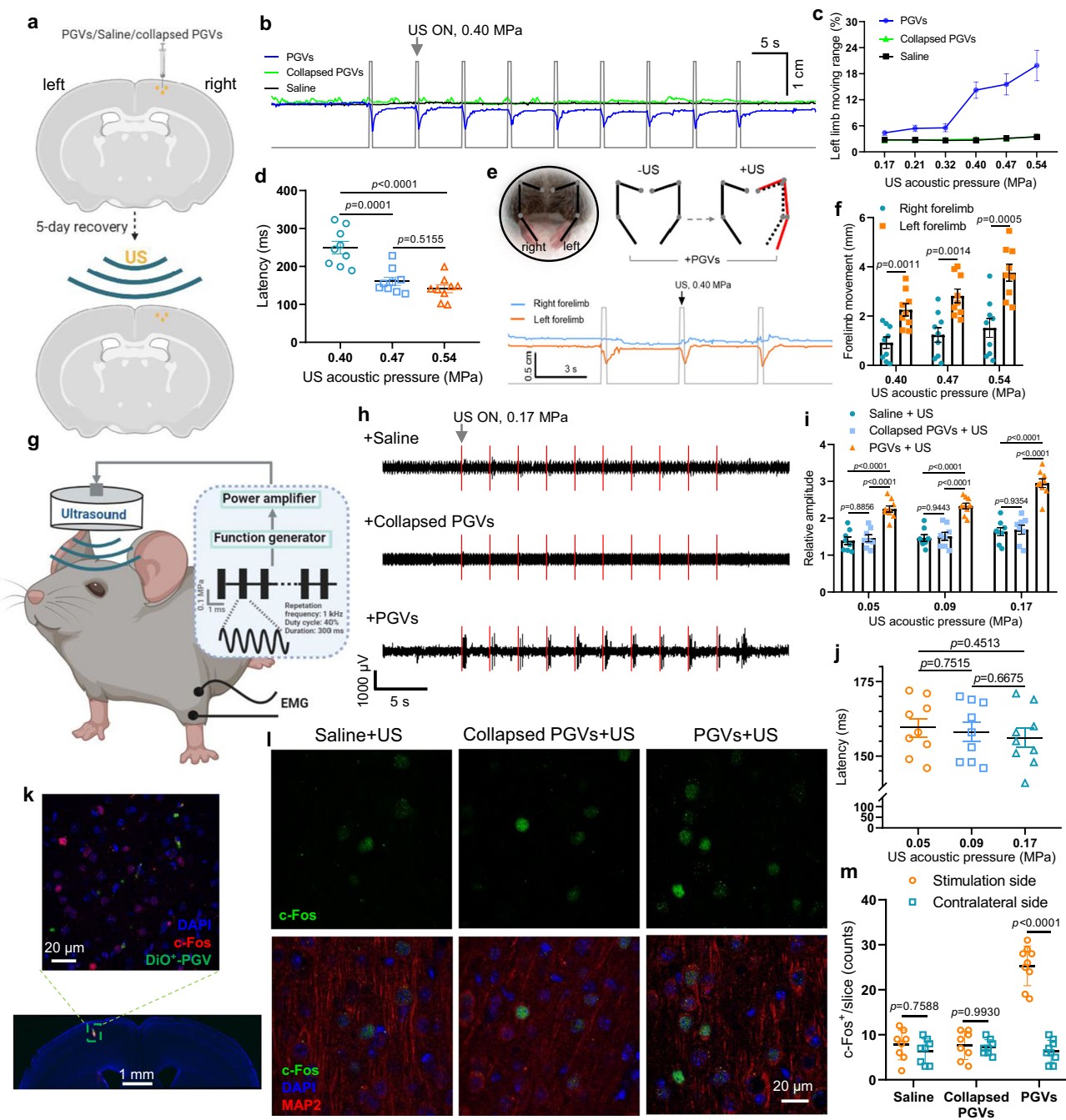

positive cells in the PGVs[+] group was over thrice that in cortices of Saline or collapsed-PGVs mice (Fig. 3l, m). Thus, we found the PGVs+US setup was an effective and targeted method of selectively activating a desired region of the mouse motor cortex, increasing neuronal activation and triggering repeatable motor responses with low latencies. No such consistent pattern of activation was seen in regions of the brain without PGVs, or with the injection of collapsed PGVs or saline. Thus, we found that PGVs could successfully lower the threshold of ultrasound needed for neuromodulation in the cortices of mouse brains.

## PGVs-actuated US can selectively stimulate different areas of the striatum to alter specific locomotor behaviors of mice

We next examined the efficiency of PGVs+US stimulation in brain regions deeper than the cortex. The striatum is known to control specific kinds of motor behavior[38], and has been targeted in other

studies to demonstrate neuromodulation. Magnetothermal and optogenetic activation of the striatum could evoke increased locomotion in the form of unilateral rotation around the body axis[39–41]. To validate our stimulation effects, PGVs were injected into the targeted (right hemisphere) dorsal striatum (3.0 mm depth), and a miniature transducer was fixed on the head for stimulation (Fig. 4a)[42,43]. A camera was employed to record the mice's movements, and the recorded videos were analyzed with the program TRACKER. It was found that during US stimulation PGVs[+] mice turned anti-clockwise around their bodies' axes, and their heads tracked circles of $3 \pm 0.5$ cm radius (Fig. 4b, Supplementary Movie 3) consistently and rapidly ($3.39 \pm 0.47$ cm/s; Fig. 4c,d), resulting in $5.61 \pm 0.79$ rev/min (Fig. 4e). The average response of PGVs[+] mice was 18-fold growth of linear speed upon ultrasound stimulation compared with the Saline[+] mice (Fig. 4d). Crucially, before US was delivered, PGVs[+] and Saline[+] mice showed no significant differences in the rotation behavior (in terms of the linear

**Fig. 3 | PGVs + US stimulation of the motor cortex triggered targeted neuronal activation, selective limb movement, and muscular responses. a** Schematic illustration of PGVs/Saline/Collapsed PGVs injection (right hemisphere), followed by a 5-day recovery period, then ultrasound stimulation. **b** Representative forelimb movements in response to 0.40 MPa ultrasound stimulation from mice injected with saline, collapsed PGVs, or PGVs. **c** Visible left forelimb movements in different groups of subjects ($n = 9$ mice for PGVs$^+$ group, $n = 8$ mice for Saline and Collapsed PGVs-injected group) were scored by percentage of movement distance in response to several acoustic intensities. ($P_{Saline/Collapsed\ PGVs} = 0.7508_{(0.17\ MPa)}$, $> 0.9999_{(0.21\ MPa)}$, $0.8271_{(0.32\ MPa)}$, $0.5293_{(0.4\ MPa)}$, $0.8234_{(0.47\ MPa)}$, $0.9747_{(0.54\ MPa)}$; ($P_{PGVs/Collapsed\ PGVs} = 0.0407_{(0.17\ MPa)}$, $0.0152_{(0.21\ MPa)}$, $0.0416_{(0.32\ MPa)}$, $0.0008_{(0.4\ MPa)}$, $0.0022_{(0.47\ MPa)}$, $0.0041_{(0.54\ MPa)}$; ($P_{PGVs/Saline} = 0.0617_{(0.17\ MPa)}$, $0.0152_{(0.21\ MPa)}$, $0.0322_{(0.32\ MPa)}$, $0.0007_{(0.4\ MPa)}$, $0.0024_{(0.47\ MPa)}$, $0.0040_{(0.54\ MPa)}$). **d** Latency between US onset (0.40 MPa, 0.47 MPa, 0.54 MPa) and induced movement ($n = 9$ mice). **e** Photographs of a PGVs-injected mouse under ultrasound stimulation. Red and black lines indicate the kinematics of forelimbs with or without US stimulation, respectively. Comparison of left and right forelimb movements in the same mice under 0.40 MPa ultrasound +PGVs stimulation. **f** Statistics of left and right forelimb movements in PGVs-injected mice ($n = 9$) in response to US excitation. **g** Schematic illustration of our setup for EMG recordings. **h** Representative EMG signals of muscular responses under ultrasound stimulation in mice injected with saline, collapsed PGVs, and PGVs (0.17 MPa). **i** Relative amplitude of EMG response treated with saline/collapsed PGVs/PGVs-mediated ultrasound at intensities (0.05, 0.09, and 0.17 MPa). $n = 8$ mice for the Saline and Collapsed PGVs-injected group; $n = 9$ mice for the PGVs$^+$ group. **j** Latency between US onsets and the induced response ($n = 9$ mice). **k** c-Fos expression (red) around PGVs injection area (green) in mouse cortex stimulated with PGVs+US. **l** Representative c-Fos-expression (green) images of mice motor cortex treated by ultrasound in the presence of saline, collapsed PGVs, or PGVs. **m** Number of c-Fos positive neurons in $200 \times 200\ \mu m$ area per slice imaged ($n = 8$ mice in Saline and Collapsed PGVs-injected group; $n = 9$ mice for PGVs$^+$ group). Data represent 8 independent experiments in **k** and **l**. Two-way ANOVA with post-hoc Tukey test in **c** and **i**, one-way ANOVA with post-hoc Tukey test for **d** and **j**, unpaired 2-tailed t-test with Holm-Sidak correction for **f**, two-way ANOVA with Sidak correction for multiple comparisons for **m**. Data in **c**, **d**, **f**, **i**, and **j** are presented as mean ± SEM. Data in **m** are presented as mean ± SD. Source data are provided as a Source Data file.

and angular speed) (Fig. 4d,e). However, a slight effect of PGVs injection alone on the mice's rotation behaviors was noted (Fig. 4c). The rotational movements elicited by PGVs+US stimulation of the striatum were repeatable in our trial using 9 mice. In addition, PGVs$^+$ mice responded quickly to the ultrasound stimulation with a mean latency time of 915.6 ms (Fig. 4f, Supplementary Movie 3), which is faster than chemogenetic and sonothermogenetic stimulation[41,44]. Thus we found that we could control specific motor behaviors by targeting PGVs+US stimulation to the dorsal striatum region.

Previous studies have shown that optogenetic stimulation of the ridge between dorsal and ventral striatum caused the voluntary locomotion of mice[39,45]. To determine the spatial accuracy of PGVs+US stimulation, we chose another behavior, freezing of gait, by targeting an even deeper striatal area (4.1 mm depth). It was found that PGVs$^+$ mice moved freely prior to US stimulation, but during US could only move their heads, not their limbs (Fig. 4g, Supplementary Movie 4). The ambulatory movements as well as head movements of PGVs$^+$ mice were reduced significantly compared to Saline$^+$ mice during ultrasound stimulation (Fig. 4h, i). The linear speed of the PGVs$^+$ mice was dramatically decreased (-2.2 fold) during US stimulation, while Saline$^+$ mice showed only a minor reduction in the same. The mean linear speed for PGVs$^+$ mice reduced from $6.17 \pm 0.48$ cm/s pre-US to $2.78 \pm 0.49$ cm/s during US, and back up to $5.51 \pm 0.46$ cm/s post-US (Fig. 4j, k). The corresponding changes in Saline$^+$ mice were minor and statistically insignificant by contrast (Pre-US: $6.67 \pm 0.57$ cm/s; during US: $6.22 \pm 0.51$ cm/s; post-US $6.63 \pm 0.69$ cm/s) (Fig. 4j, k). In our experiments, PGVs-injected mice exhibited reliable behavioral changes during ultrasound stimulation, and the affected behaviors differed depending on the brain region that was targeted. This targeting was effectively achieved by simply injecting the PGVs in different locations. Further, PGVs+US mice showed significant recovery to baseline behavior patterns in the post-US period, indicating the specificity and efficacy of the treatments. Therefore, we found our PGVs+US system to be effective enough to reliably excite neurons in specific deep brain regions and activate neural pathways in vivo.

## PGVs-actuated US can selectively activate neurons in targeted regions of the striatum

To further examine the mechanism of the behavior changes that the PGVs+US stimulation was able to trigger, we monitored the activity of neurons in the mouse dorsal striatum through fiber photometry. Neurons in this region were transduced with recombinant adeno-associated virus (rAAV) vectors to express the genetically-encoded calcium sensor jRGECO1, and an optical fiber was inserted to record neuronal calcium fluxes during stimulation (Fig. 5a, b). We discovered that in the presence of PGVs, jRGECO1a fluorescence increased repeatedly, rapidly, and reversibly even under extremely low-intensity ultrasound stimuli (0.09 MPa), but not in Saline$^+$ mice (Fig. 5c, d). In addition, the PGVs+US-evoked calcium responses were ultrasound intensity-dependent, but the Saline$^+$ group showed essentially no responses even under the highest acoustic pressure tested (0.21 MPa) (Fig. 5e). Notably, there were no repetitive patterns of fluorescence changes before ultrasound onsets. The applied ultrasound parameters did induce some degree of Ca$^{2+}$ responses in the Saline$^+$ mice, but the average amplitude in the PGVs+US group was significantly higher under all applied US intensities (Fig. 5e). Response latency was found to be 233.3 ms, 217.7 ms, and 203.3 ms for 0.09 MPa, 0.17 MPa, and 0.21 MPa, respectively, suggesting that the elicited PGVs+US calcium responses were dependent upon acoustic intensity (Fig. 5f). We keep monitoring the neural Ca$^{2+}$ responses via fiber photometry for more days and found that the PGVs+US-elicited Ca$^{2+}$ signals persisted until day 12, indicating that PGVs' effective time was up to 12 days in Supplementary Fig. 6a, b. Such results are in line with our PGVs imaging data (Fig. 2b–d).

Additionally, we confirmed the specific activation of neurons in the specific areas of the striatum that were targeted to elicit different behaviors - 3.0 mm depth for rotation, and 4.1 mm depth for freezing respectively - by immunohistological staining of c-Fos. The number of c-Fos$^+$ neurons was dramatically greater in PGVs$^+$ mice following the US, but not in any of the other groups (Fig. 5g, h, Supplementary Fig. 7a, b). An increase in c-Fos$^+$ cells was detected in the specific PGVs$^+$ striatal region corresponding to the elicited behavior but not in the other striatal region, and not in the auditory cortex (Fig. 5i, Supplementary Fig. 7a–c). We also confirmed that most of the c-Fos$^+$ cells were neurons by MAP2 staining (Supplementary Fig. 8). These data show that neural excitation was only activated by ultrasound in a localized manner, in the presence of PGVs. Thus, the PGVs+US scheme has robust targeting capabilities and good spatial precision, with no obvious involvement of the previously reported collateral auditory confounding effect[46,47].

## PGVs-actuated ultrasound stimulation alleviated depression-like behaviors in mice

Given the robust neuromodulation we could trigger using PGVs+US, we next tested whether the scheme could be used to exert therapeutic effects. Depression is associated with brain dysfunction resulting from neural circuit disorders[48]. Ultrasound stimulation is considered a promising treatment for various neuropsychiatric disorders, and we were thus interested in whether our PGVs+US neuromodulation strategy could affect this kind of condition. Thus, we chose depression as our targeted disease model. First, a model of depression-like mice was established through chronic restraint stress (Fig. 6a)[49]. These mice

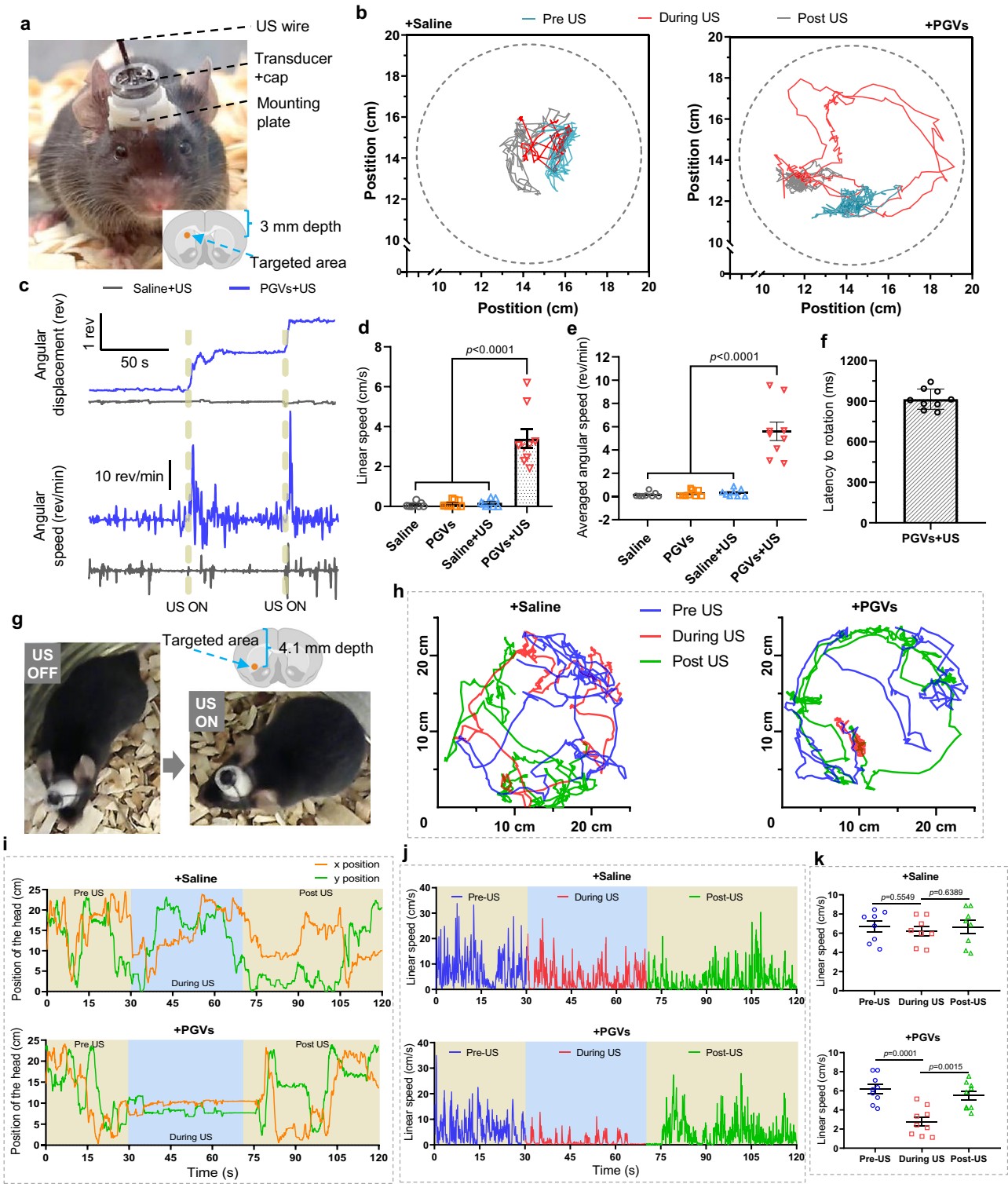

showed decreased body weight and increased immobility time after the 14-day repeated confinement (3 h/day), indicating the successful establishment of the depression model (Supplementary Fig. 9a–d). The DRN is the major serotonergic nucleus in the brain, and it offers massive projections to different brain areas[50]. Previous studies have demonstrated that activation of serotonergic neurons in the DRN is sufficient to induce an antidepressant-like effect in mice[51]. Therefore, we chose it as our targeted brain region to treat depression-like mice with PGVs+US (Supplementary Fig. 9a). DRN neurons were transduced with rAAV vectors to express a fluorescent serotonin (5-HT) sensor

(Fig. 6b). PGVs+US stimulation triggered repeatable 5-HT release from DRN neurons, and the response amplitude increased progressively with the ultrasound intensity, but not in the Saline+ group (Fig. 6c–e). We also measured the fluorescence dynamics in response to PGVs+US stimulation and detected 114.4 ms (0.13 MPa), 108.9 ms (0.25 MPa), and 103.3 ms (0.30 MPa) latency (Fig. 6f). These findings suggest that PGVs+US stimulation could evoke endogenous 5-HT release in the DRN with excellent temporal precision. To test the stimulation-induced serotonergic activity, we checked c-Fos in 5-HT neurons in this region to show serotonergic neurons activation, which can reduce

**Fig. 4 | PGVs + US stimulation of targeted deep brain regions alters specific mouse locomotor behaviors (rotation, freezing). a** Ultrasound stimulation setup on the head of a mouse. A wearable ultrasound transducer was placed at the mounting plate stuck to the mouse skull. Schematic illustration of the brain indicates the injection site in the dorsal striatum. **b** Typical trajectories showing the movements of Saline[+] (left) or PGVs[+] (right) mouse (post US: 27–47 s was not included). **c** (Top) A plot of the angular displacement showing a mouse turning during PGVs+US stimulation. Counter-clockwise angular changes were counted as positive changes in angles. During US onsets, the mouse turned unilaterally significantly more than pre-US. (Bottom) A trace shows the angular speeds of the mouse. Speed markedly increases during US onsets (dotted lines). **d, e** The average linear and angular speeds of PGVs[+] mice, and Saline[+] mice with or without 0.56 MPa ultrasound. $n = 8$ mice for Saline and Saline+US group, $n = 9$ mice for PGVs and PGVs+US group. **f** The latency of behavior onset in response to PGVs+US stimulation ($n = 9$ mice). **g** Still images of the same PGVs[+] mouse before (left) and during (right) the ultrasound stimulation. The mouse could not execute limb movements under ultrasound stimulation. In the absence of ultrasound, the ambulation was back to normal. Schematic illustration of the brain indicates a different injection site, on the ridge between the dorsal and ventral striatum. **h** Representative trajectories recorded of mice pre-, during, and post-US. **i** Recorded head positions from 1 representative Saline[+] and PGVs[+] mouse each over the observed period (X and Y). **j** Linear speed of 1 representative PGVs[+] and Saline[+] mouse each under 0.56 MPa ultrasound treatment during the observed period (1 kHz PRF, 3 s burst interval). **k** Average recorded linear speeds of Saline[+] or PGVs[+] mice. Mice $n = 8$ for Saline and Saline+US group, $n = 9$ for PGVs and PGVs+US group. One-way ANOVA with post-hoc Tukey test for **d, e** and **k**. Data in **d, e**, and **k** are presented as mean ± SEM. Data in **f** are presented as mean ± SD. Source data are provided as a Source Data file.

depression-like behavior. We found a 2.5-fold increased expression of c-Fos in 5-HT[+] neurons in the PGVs+US condition compared to the Saline[+] condition (Supplementary Fig. 9e, f). To further evaluate the proportion of activated DRN 5-HT neurons, we examined the expression of TPH2, a representative marker of serotonergic neurons[52] (5-HT neurons) and c-Fos simultaneously. We found a significantly higher percentage of c-Fos expression in TPH2-positive neurons in the PGVs +US group (73.2%) than the others (Saline, 13.7%; PGVs, 15.3%; Saline +US, 17.3%, Fig. 6g, h). Altogether, these data indicate that PGVs+US could successfully excite serotonergic neuron cells in mice DRN with high spatial and temporal precision.

Most antidepressants are effective in reversing this resting state[53]. The tail suspension test (TST) and forced swimming test (FST) are well-established methods of evaluating the depressive state of mice and assessing the efficacy of antidepressants[54]. These tests treat a mouse's stillness when in discomfort as a substitute for a sense of helplessness, and an increase in a mouse's tendency to move as a reduction in their depression-like behavior. Hence, we applied these two tests to observe if PGVs+US stimulation of the serotonergic neurons in the DRN could attenuate the depression-like behavior of mice. Only the PGVs+US group displayed significantly decreased immobility time, while Saline +US, untreated, PGVs[+]-only and Saline[+]-only mice showed comparable immobility times (Fig. 6i, k, Supplementary Movie 5 & 6). Similarly, we also found that PGVs+US mice showed a longer struggling time than mice in other groups after ultrasound stimulation (Fig. 6j, l). These results suggest that PGVs+US stimulation of DRN can promote active coping with stress and decrease depression-like symptoms in mice. These behavior tests demonstrate that PGVs-actuated US enables precise and targeted activation of the DRN serotonergic neurons, which can rapidly promote active stress-coping in mice and effectuate an improvement in mice models of depression.

## Biosafety assessment of PGVs with/without ultrasound stimulation

We have assessed the cellular safety of PGVs and found that the percentages of cell viability at each incubation time point were higher than 94.3%, indicating the low cytotoxicity of the PGVs (Fig. 1f). Furthermore, the mice had steady body weights, and none died in the post-injection period, indicating no obvious systemic toxicity (Fig. 7a). Given the PGVs stay for a period in the mouse brain, it is important to verify that they do not change the normal function of the brain. Therefore, we monitored the mice behaviors after PGVs treatment and found that PGVs did not affect general activities such as locomotor activity (Fig. 7b, c), memory (Fig. 7d, e, Supplementary Fig. 10a), or cognition (Fig. 7f–h, Supplementary Fig. 10b, c) comparing with Saline[+] mice group. To further identify the biosafety of PGVs, we examined the expression of the specific markers on brain tissues after PGVs injection with/without ultrasound stimulation. Immunohistology examination of mice brains after our treatment did not find any significant changes in the percentages of microglia, astrocytes, and apoptotic cells

compared with the control, suggesting good biosafety of our stimulation strategy (Fig. 7i, j). Our results suggest that PGVs are safe mediators both in vitro and in vivo, which are promising ultrasound actuators for precise neuromodulation.

## Discussion

Non-invasive neuromodulation technologies with high spatiotemporal resolution hold great potential for advancing brain research in many directions. Here, we employed a low-frequency transducer and demonstrated that PGVs can help to localize acoustic energy to decrease the threshold of ultrasound intensity for neurostimulation without the need for genetic modification. We achieved controllable calcium signaling in primary neurons via the activation of mechanosensitive ion channels. A scheme like ours may be broadly applied to identify cells that show increased mechanosensitivity than surrounding tissue, such as in aging brains or the progression of cancers[55,56]. The ease of the GVs' surface modification could also enable cellular targeting through attached ligands, or other targeting molecules[57,58]. Thus, the ability of PGVs-mediated ultrasound stimulation to act upon mechanosensitive ion channels could be applied in fields beyond neurostimulation.

Our previous study showed that sonogenetics could activate mice's motor cortex and trigger EMG signals using low ultrasound intensities. In contrast, PGVs+US evoked greater EMG responses at the same acoustic pressure, indicating better amplification of acoustic effects by PGVs[19]. While not a direct, apples-to-apples comparison, given the different experimental setups, we believe that the nanobubbles-enabled ultrasound is promising as it does not require genetic modification. Moreover, our strategy is for localized enhancement or amplification of ultrasound that does not exclude sonogenetics for cell-type selective stimulation, and could even be combined with it. Given that the neuromodulatory effect of PGVs+US is localized mainly through the presence of nanobubbles, this approach also allows us to get around the inherent diffraction-limitation of ultrasound waves without imaging guidance.

In our experiments, PGVs+US activated targeted brain areas, including deep-seated regions, with high spatial resolution and induced specific behavior alterations in mice. We successfully triggered defined behaviors, rotation or freezing, by targeting the dorsal striatum or the ridge between the dorsal and ventral striatum, correspondingly, which are only ~1.1 mm apart in a mouse brain. Thus, we match the neuromodulation capabilities previously demonstrated with optogenetic[39,59], chemogenetic[60], and magnetothermal[40] approaches. The improved spatial resolution of neural activation may come from the different experimental procedures between this study and our earlier work[18]. In the present study, mice were allowed a 5-day recovery period after PGVs injection, whereas c-Fos data in the previous work were from mice treated shortly after GVs were delivered into the brain. Given the disparity in post-injection times, some PGVs may have been degraded and cleared from peripheral areas in the

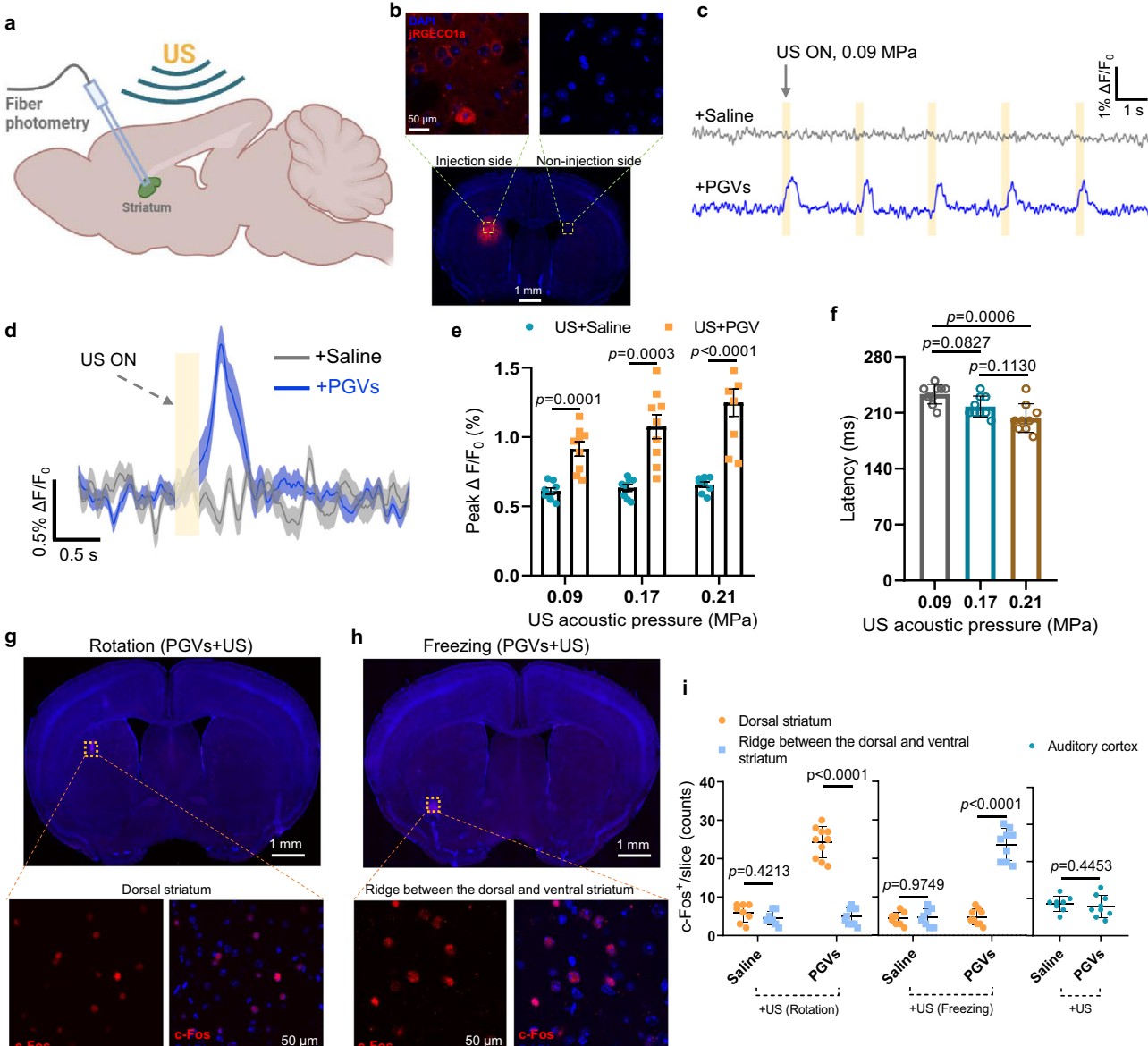

**Fig. 5 | Targeted neuronal excitation in vivo in the mouse striatum. a** Schematic illustrating the fiber photometry setup for measuring Ca$^{2+}$ dynamics stimulation in the striatum. **b** Representative images of jRGECO1a expression in neurons of the dorsal striatum. **c** Representative jRGECO1a fluorescent traces from the striatum of Saline$^{+}$ or PGVs$^{+}$ mice under US stimulation (0.09 MPa peak pressure). **d** Mean jRGECO1a fluorescence traces. The mean trace is solid and SD is shaded. **e** Average peak of Ca$^{2+}$ responses ($\Delta F/F_0$) in response to 0.09, 0.17, 0.21 MPa ultrasound stimulation. Mice $n = 8$ in Saline+US group, $n = 9$ for PGVs+US group. **f** Latency between US onset and induced Ca$^{2+}$ response ($n = 9$ mice). **g**, **h** Typical images of mouse brain slices with c-Fos staining. (Top) Whole-brain slice images showing

c-Fos expressing. (Bottom) Enlarged areas of Striatum with c-Fos staining. **i** c-Fos counts in slices of targeted brain regions and non-targeted regions ($n = 8$ for the Saline+US group and $n = 10$ for the PGVs+US group in the Rotation test, $n = 8$ for the Saline+US group and $n = 9$ for PGVs+US group in Freezing test, $n = 8$ for Saline+US group and $n = 9$ for PGVs+US group in Auditory cortex). Images represent 8 independent experiments in panels **b**, **g** and **h**. Two-tailed unpaired t-test with Holm–Sidak correction in **e**, one-way ANOVA with post-hoc Tukey test in **f**, two-way ANOVA with Sidak correction for multiple comparisons in **i**. Data in **e** are presented as mean ± SEM. Data in **f** and **i** are presented as mean ± SD. Source data are provided as a Source Data file.

intervening period, making the impacted area smaller and more localized. We also demonstrated the expected neuronal activation and calcium dynamics directly in the areas targeted to trigger the behavior, and not in others. Taken together, these results show that the evoked behaviors resulted from the activation of specific neural circuits and that they were in response to the localized mechanical energy transmitted by the PGVs.

Our ultrasound stimulation scheme resulted in rapid induction of targeted motor behavior, which was observable within 1 s of activating the ultrasonic stimulus. It is important to note that this brief delay is not inherent to the technique itself but rather a result of the

time required for the entire neural circuits to adapt and produce the desired behavioral change. It is, thus, feasible to reduce the latency of behavioral responses to ultrasound to times which are only marginally slower than electrophysiological or optogenetic stimulation, and significantly faster than chemical stimulation using DREADDs[44]. Likewise, our results show that the evoked locomotor behavior diminished quickly within 5 s after the ultrasound was turned off. Accurate temporal control, both for stimulation and cessation, is crucial to establish precise correlations between circuit activation and behavioral observations. So far, achieving such precise control has been primarily demonstrated through optogenetics, while

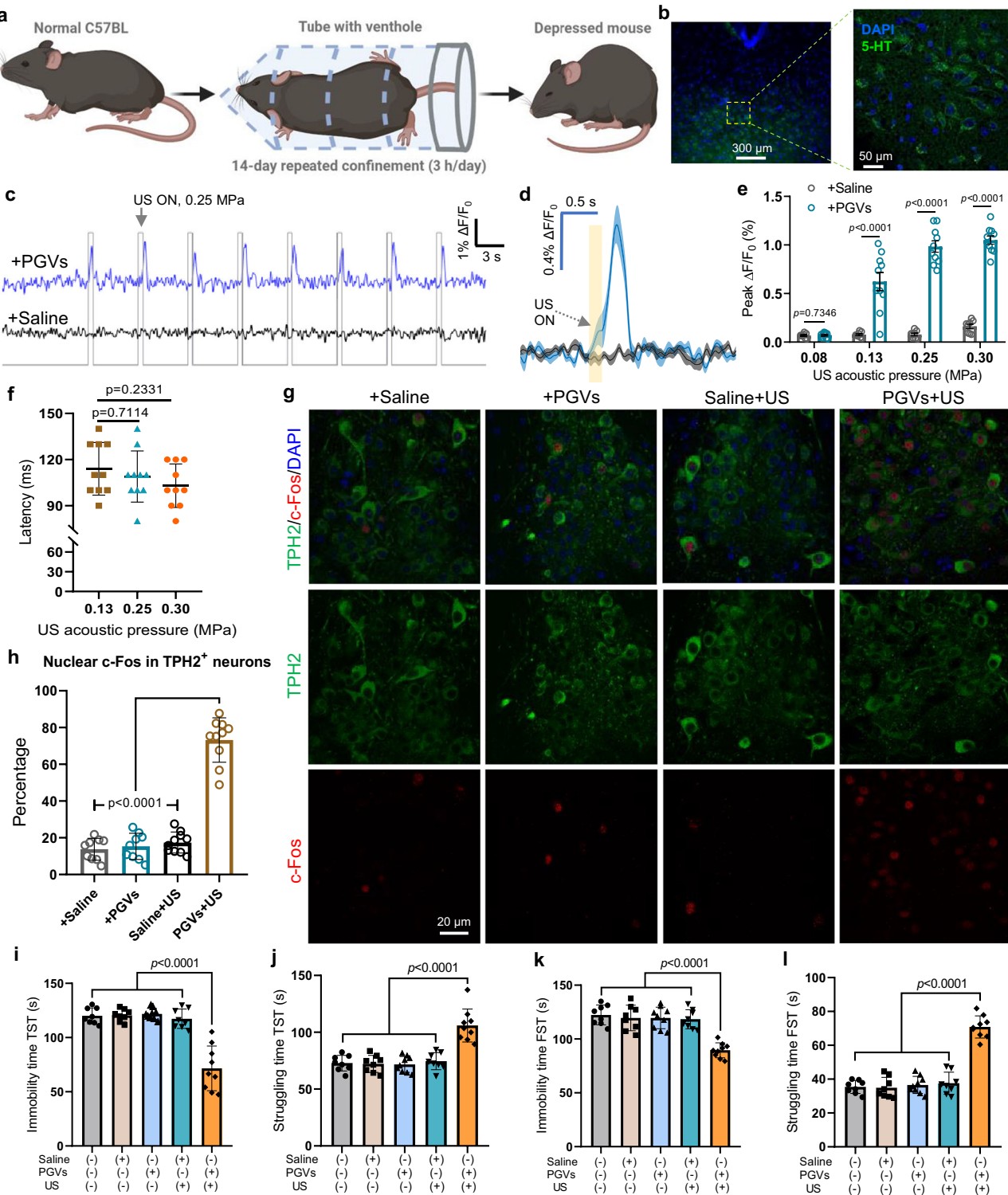

magneto- or chemogenetics have not been able to match this level of temporal precision[40].

A potential concern with our scheme is the possibility of cell death induced by the nanobubbles, either through direct effects or membrane sonoporation. However, we addressed this concern by conducting multiple trials of stimulation on the same animals over a span of a few days. Remarkably, we observed no alterations in normal or evoked behavior, suggesting that the nanobubbles were well-tolerated without inducing any detectable cell damage. Additionally, immunohistochemical staining of brain slices showed negligible inflammation and apoptosis after the stimulation. Hence, we did not detect any

damage using the standard methods of weight tracking, and histological and behavioral monitoring. It is also known that mechanical and thermal damage can be avoided by choosing a low ultrasound intensity to decrease the possible inertial cavitation and temperature rise. In our experiments, we took care to restrict ourselves to the lower end of ultrasound intensities for this exact reason, with the highest applied intensity being 0.56 MPa.

A limitation of this study is that it lacks information about ion-channel dynamics, which could be obtained by patch clamping during ultrasound stimulation. We found this to be unachievable during the present study as we could not eliminate the vibration of the patch

**Fig. 6 | PGVs + US stimulation-induced neural activation in the DRN and reduction of depression-like behavior. a** Schematic illustration of establishing a depression mouse model through chronic restraint stress (CRS) for 14 days. **b** Representative confocal images of 5-HT sensor expression (green) in the mouse DRN. **c** Representative 5-HT fluorescence responses in the DRN of anesthetized PGVs[+] or Saline[+] mice to US stimulation (0.25 MPa). **d** Averaged 5-HT sensor signals in response to US+Saline/PGVs stimulation. The light-yellow line represents the US onset. The mean trace is solid and SD is shaded. **e** Average peak 5-HT response in Saline[+] mice ($n = 8$) or PGVs[+] mice ($n = 10$). **f** Latency between the US onsets at varying intensities and induced 5-HT response ($n = 10$ mice). **g** Representative immunofluorescence images of c-Fos (red) / TPH2 (green) in the DRN treated with the indicated conditions. **h** Calculated c-Fos-positive nuclei in TPH2[+] neurons under different treatment conditions ($n = 9$ mice for Saline and PGVs group, $n = 10$ mice for Saline+US and PGVs+US group). **i** Graph showing the immobility time of depression-like mice during the tail suspension test (TST). **j** Graph displaying the struggling time of depression-like mice during the TST. **k** Graph showing the immobility time of depression-like mice during the forced swimming test (FST). **l** Graph displaying the struggling time of depression-like mice during the FST. Mice number, $n = 8$ for control, Saline, and Saline+US group, $n = 9$ for PGVs and PGVs+US group in panels **i**–**l**. Two-tailed unpaired t-test with Holm−Sidak correction in **e**, one-way ANOVA with post-hoc Tukey test for panels **f**, **h**–**l**. Images represent 8 and 9 independent experiments in panels **b** and **g**, separately. Data in (**e**) are presented as mean ± SEM. Data in **f**, **h**–**l** are presented as mean ± SD. Source data are provided as a Source Data file.

pipettes when ultrasound was turned on, which has been discussed in other studies as well[61,62]. Indeed, it would be useful to know the specific parameters required to trigger channel opening in PGVs-mediated ultrasound stimulation. However, we were able to collect evidence of the role of mechanosensitive ion channels through calcium imaging by using various blockers, and collected real-time evidence of in vivo calcium and serotonin dynamics. This gives us a useful indicator of the neuronal dynamics involved, even though implementing the gold-standard technique has been infeasible so far.

Similar to genetically-based and other neuromodulation techniques, a major challenge for the application of our method in humans will be addressing safety concerns associated with the injection of viruses and nanoparticles into the brain. However, extensive research has been conducted on the body's response to nanoparticle injections, and this reaction can be mitigated by utilizing nanoparticle surface coatings that are biologically compatible. It is worth noting that the invasive brain surgery currently used for GVs injection may soon be substituted by a cellular expression of GVs or delivery by blood-brain barrier (BBB) opening using focused, non-invasive ultrasound[63]. Overall, the modular nature of nanoparticles like GVs and their non-invasiveness offer many choices and advantages for brain stimulation.

## Methods

### Gas vesicle preparation and modification

*Anabaena flos-aquae* was cultured in sterile BG-11 medium at 25 °C under illumination with a 14/10 h light/dark cycle. GVs were extracted by hypertonic lysis to release GVs by quickly adding sucrose solution to a final concentration of 25%. Following lysis, GVs were separated by centrifugation at $400 \times g$ for a duration of 3 h. For GVs purification, the resultant solution underwent trine centrifugation cycles and was subsequently preserved in PBS at 4 °C. The GVs' concentration was measured by optical density at 500 nm ($OD_{500}$) utilizing a UV-Visible spectrophotometer[64,65]. For PEGylated GVs (PGVs) synthesis, PEG was adsorbed on the GVs protein shell by covalent coupling. First, EDC/NHS was added into a medium of PEG-amine in PBS[57,66]. It was then stirred at 25 °C for 2 h. Then GVs were added dropwise into the pre-mixed PBS solution. Such reaction compound was stirred at 4 °C for an additional 24 h (Supplementary Fig. 2a). PEG was coupled to GVs via peptide bond formation in the company of EDC/NHS. The resulting mixture was transferred into the 2 mL tube and washed 4 times with PBS to eliminate the excess amount of chemicals. The modified nanobubbles were suspended in a PBS buffer at 4 °C for further study.

The GVs and PGVs were marked with the near-infrared dye, Indocyanine green (ICG), for mouse brain imaging tests. In brief, EDC/NHS was put into a medium of ICG in PBS. This solution was added to the GVs/PGVs solution after a 30 min incubation at 25 °C (ICG: GV/PGV = 1000:1). It was then shaken for 24 h at 4 °C and then washed 4 times through centrifugation. The resulting mixture was transferred into another 2 mL tube followed by 4 additional rounds of centrifugation for 5 min to remove free ICG. The modified nanobubbles were suspended in PBS and protected from light.

### Acoustic field characterization and passive cavitation detection

A flat transducer with a center frequency of 1.0 MHz (A303S, Olympus) was utilized in our investigation. Ultrasonic pulses were generated through a function generator (AFG251, Tektronix) and a power amplifier (A075, Electronics & Innovation Ltd.). The transducer being tested was immersed in deionized, degassed water in a large tank, together with a hydrophone (Supplementary Fig. 4b). The acoustic intensity profile was characterized by a hydrophone. Point-by-point scanning was done by computer-controlled 3-axis translation stage.

A hollow chamber made of 3% agar was filled with sample suspension and placed in a large tank filled with deionized, degassed water. Transducer and hydrophone (HGL-0200, Onda) were also immersed and aligned at 90°, both pointing at the center of the chamber (Supplementary Fig. 4c). Repeats of recorded signal (200 μs) were treated with fast Fourier transform (FFT) and resulting power spectra were averaged.

### Preparation of PA hydrogels for neuron culture

The confocal dishes were coated with a 1.0 kPa polyacrylamide (PA) gel with stiffness much lower than that of glass[67]. Briefly, Confocal dishes were rinsed with 0.1 M NaOH, dried, incubated for 5 min with 3-aminopropyltriethoxysilane, washed with distilled water, incubated with 0.5% glutaraldehyde for 30 min, and air-dried. The PA-gel solution was then added to the dishes such that the gel's final thickness would be around 50 μm, and a diameter of 12 mm coverslip was gently laid on it. The solution contained APS, TEMED, acrylamide, and bis-acrylamide (all from Pierce Biotechnology) at a ratio such that the polymerized gel would have a stiffness of 1.0 kPa. The gels were allowed to polymerize, the coverslips removed, and the dishes were thrice washed with PBS. The gels were then functionalized by adding a sulfo-SANPAH (Pierce Biotechnology) solution and exposing them to UV light of 365 nm for 15 minutes to covalently link the gel to the sulfo-SANPAH. Next, the dishes were sterilized by UV light in a cell culture hood for 30 min and then coated with poly-L-lysine (PLL, Gibco) for the following cell seeding.

### Cell culture

All cells were grown inside a standard humidified cell culture incubator at 37 °C with 5% $CO_2$. Neurons from rat embryos at embryonic day 18 were obtained[68]. In summary, cortical tissues were meticulously dissected and subjected to a 0.25% trypsin treatment for a duration of 15 min at 37 °C, accompanied by gentle agitation. Termination of the enzymatic digestion was achieved by introducing Neurobasal medium (Gibco) with 10% fetal bovine serum and 1% penicillin-streptomycin. The cells were resuspended in a medium and gently mechanically triturated with a pipette, followed by a 15-minute incubation period. The resultant supernatant was discarded, and the cells were resuspended in the aforementioned medium before being plated at a density of $1 \times 10^5$ cells/cm² in confocal dishes with PLL-coated 1.0 kPa PA-gel. Subsequently, after a 24-hour incubation period, the medium was changed to Neurobasal + 2% B27 + 0.25% L-Glutamine + 1% Penicillin-

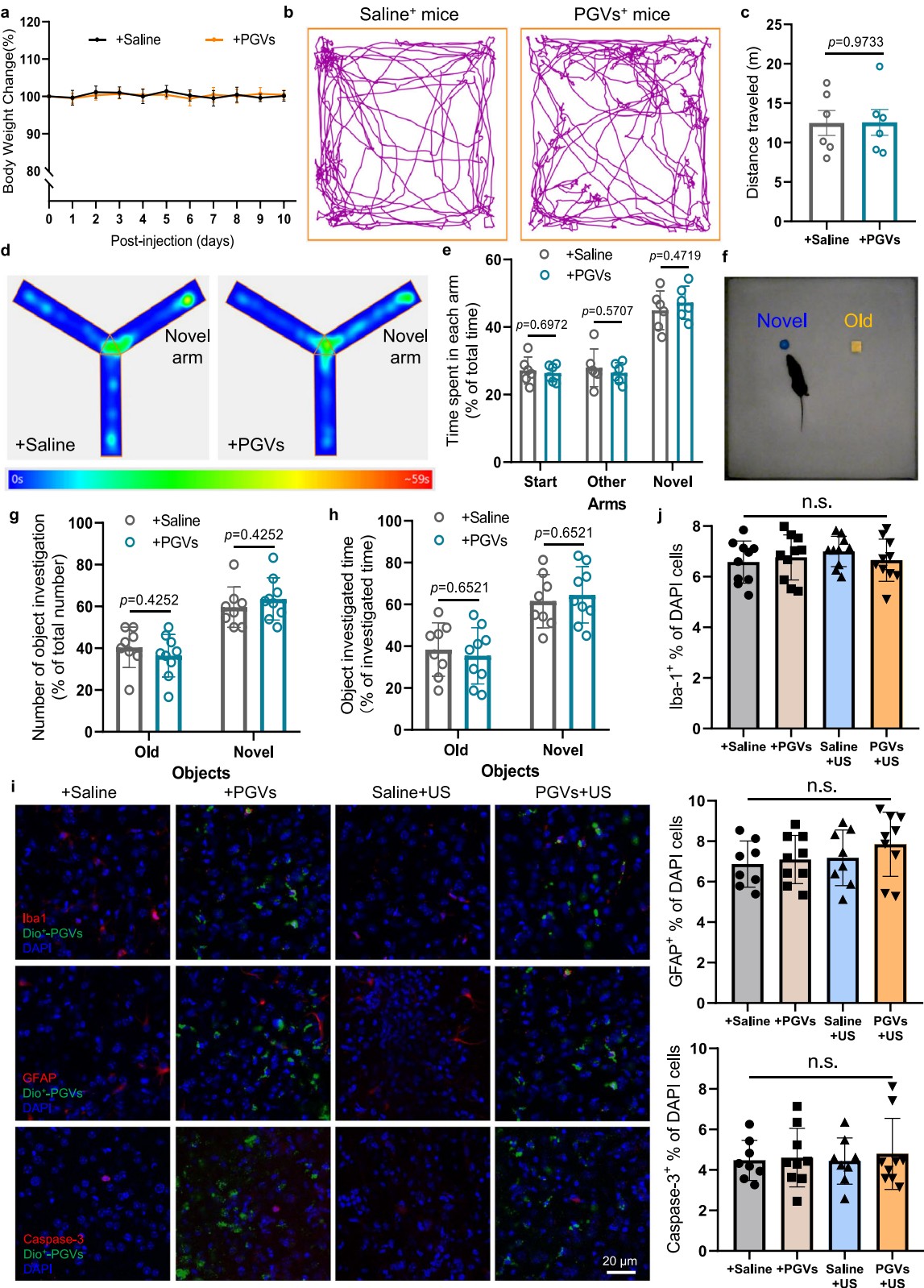

Streptomycin (all from Gibco). Thereafter, half of the medium was refreshed every 2-3 days.

## In vitro ultrasound stimulation

For the present study, we used a customized system that facilitated ultrasound stimulation and calcium imaging simultaneously (Supplementary Fig. 3a). Concisely, the ultrasound stimulation system was aligned with a calcium imaging system to monitor the calcium responses of the stimulated neurons. Ultrasound waves were delivered through a waveguide containing degassed water that was attached to the ultrasound transducer assembly. Cells were cultured on PA-gel inside a confocal dish, and PGVs were added to the medium and gently mixed just before stimulation. Prior to cellular stimulation, the acoustic pressure and field characteristics of the system

**Fig. 7 | Biosafety assessment of PGVs. a** Body weight measurement of mice during the 10-day evaluation period (*n* = 9 mice in the Saline⁺ group, *n* = 10 mice in the PGVs⁺ group). **b** Representative trajectories recorded from mice injected with Saline or PGVs into dorsal striatum (each trace 5 min long). No side effects were observed after PGVs administration. **c** Total distance traveled in open field test (OFT) 5 days after injection of PGVs or Saline. *n* = 6 mice in each group. **d** Heatmap of the mice movement in Y maze. **e** Calculated percentage of time spent by mice on the three arms (start, other, novel areas) in Saline⁺ and PGVs⁺ mice. *n* = 6 mice in each group. **f** Images of the mouse in the novel object recognition (NOR) field. **g** Percentage of number of object investigation by saline⁺ mice (*n* = 8) or PGVs⁺ mice (*n* = 9). **h** Calculated percentage of time spent by mice on the old or novel objects. *n* = 8 mice in the saline group, *n* = 9 mice in the PGVs group. **i** Evaluation of neural inflammation, and apoptosis after US exposure in the US-treated brain region using immunohistochemical staining of microglia (Iba1), astrocytes (GFAP), and Caspase-3, green shows the injected PGVs, blue indicates the DAPI-stained nuclei. Images represent 8 independent experiments. **j** Counts of cells in 450 × 450 µm area per slice imaged. Positively stained for the 3 indicated proteins as a percentage of DAPI-stained cells in the same field of view, within the targeted brain region in mice with and without US treatment. *n* = 8 mice in the Saline, Saline+US group, *n* = 9 mice in the PGVs, PGVs+US group. *p* > 0.05 ($P_{PGVs/Saline}$ = 0.9548$_{(Iba-1)}$, 0.9852$_{(GFAP)}$, 0.9970$_{(Caspase-3)}$; $P_{Saline+US/Saline}$ = 0.6452$_{(Iba-1)}$, 0.9672$_{(GFAP)}$, > 0.9999$_{(Caspase-3)}$; $P_{PGVs+US/Saline+US}$ = 0.7629$_{(Iba-1)}$, 0.7332$_{(GFAP)}$, 0.9508$_{(Caspase-3)}$; $P_{PGVs+US/PGVs}$ = 0.9888$_{(Iba-1)}$, 0.6374$_{(GFAP)}$, 0.9915$_{(Caspase-3)}$) no significant differences (n.s.). The two-tailed unpaired t-test in **c**, two-tailed unpaired t-test with Holm–Sidak correction in **e**, **g**, and **h**, one-way ANOVA with post-hoc Tukey test in **j**. Data in **a**, **e**, **g**, **h** and **j** are presented as mean ± SD. Data in **c** are presented as mean ± SEM. Source data are provided as a Source Data file.

were assessed using a hydrophone, revealing a consistently homogeneous ultrasound field in the central region (Supplementary Fig. 4a). Each stimulus was composed of 300-tone burst pulses at a center frequency of 1.0 MHz, 10% duty cycle, pulse repetition frequency (PRF) of 1 kHz, at low acoustic intensity (0.20 MPa). These parameters ensured the delivery of ultrasound in short bursts, thereby minimizing thermal effects. For experiments not requiring real-time imaging, cellular treatments were conducted within a standard cell culture incubator[19].

## Calcium imaging

The culture medium was replaced with a working solution of Fluo-4 AM (5.0 µM) (Invitrogen) in Ca²⁺ solution (pH 7.4), and the cells were incubated at 37 °C in darkness for 30 min. Following incubation, a fresh Ca²⁺ solution was used to flush away excess dye before ultrasound stimulation. Calcium imaging was conducted utilizing a modified inverted fluorescence microscope. The excitation light was generated by a dual-color LED, filtered and delivered to the sample to illuminate the calcium sensor. To minimize phototoxicity, the LEDs were triggered at 1 Hz (except Supplementary Fig. 3b which used 2 Hz) and synchronized with sCMOS time-lapse imaging. Confocal dishes with dye-loaded cells were placed above the objective, and PGVs were distributed into the media directly before ultrasound stimulation. Intracellular images of Fluo-4 AM were recorded using a camera at defined time intervals with excitation wavelengths of 494 nm for Fluo-4 AM. We used software (cellSens) to communicate and coordinate the operation sequence between the microscope and the monochromator. The cultured neuronal Ca²⁺ responses were processed with ImageJ 1.52c and MATLAB R2020b with our custom code in Supplementary Information.

## MTT assay

MTT assays were used to evaluate the cytotoxicity of PGVs in the cultured neuron cells. Neurons were subjected to PGVs treatments in 96-well plates. Following the designated treatments and incubation periods, cells were exposed to 0.5 mg/mL MTT in the medium for 3-4 h at 37 °C. Subsequently, the cells were solubilized with DMSO and subjected to 15 min of agitation, after which the absorbance of the resulting solution at 570 nm was measured using an LEDTect 96 microplate reader[69,70].

## Animal care

Male C57BL/6 were purchased from Jackson Laboratories, and housed under standard housing conditions, with food and water available *ad libitum*. The mice were group-housed on a 12 h light: 12 h dark cycle (temperature: 20–25 °C, humidity: 50–65%). Animals from the above-mentioned groups were assigned randomly to treatment groups. All animal experiments were approved by the Animal Subjects Ethics Sub-Committee (ASESC) of the Hong Kong Polytechnic University and were performed in compliance with the guidelines of the Department of Health · Animals (Control of Experiments) of the Hong Kong S.A.R. government.

## Stereotactic injection of virus and PGVs

C57BL/6 mice at 8 weeks were anesthetized with Ketamine and Xylazine (100 mg/kg and 10 mg/kg respectively) via an intraperitoneal (IP) injection. The anesthetized mice were positioned in the stereotaxic apparatus, and ointment was applied to the eyes. Skin incisions were then performed to expose the skull. 0.5 µL AAV9-hSyn:jRGECO1a (1 × 10¹² v.g., (BrainVTA (Wuhan) Co. Ltd)) was injected into the dorsal striatum, or the 0.5 µL rAAVs-hSyn-5-HT2.1 (1 × 10¹² v.g., (BrainVTA (Wuhan) Co. Ltd)) was injected into the dorsal raphe nucleus (DRN) at 0.05 µL/min, followed by a 10-min pause. The pipette was then slowly withdrawn. After finishing the injection, skin tissue was sutured and disinfected, and mice were allowed to recover on a heating pad. Mice were then returned to their housing areas.

Targeted brain regions received 1.0 µL of PGVs (8.0 nM in saline) or saline at 0.1 µL/min by Stereotactic injection, followed by a 10-minute pause. The puncture site was then disinfected and sutured, and the mice were returned to their housing areas for recovery. The coordinates used for the motor cortex (limb movement) were AP: −0.55 mm, ML: −1.47 mm, DV: −0.55 mm; motor cortex (EMG) were AP: 0.25 mm, ML: −1.50 mm, DV: −1.00 mm; the dorsal striatum region were AP: 0 mm, ML: −2.30 mm, DV: −3.0 mm; the ridge between dorsal and ventral striatum were AP: 0.01 mm, ML: −2.30 mm, DV: −4.1 mm; and the dorsal raphe nucleus were AP: −4.0 mm, ML: 0 mm, DV: −3.4 mm.

## Nanobubbles stability and biosafety test in vivo

GVs and PGVs were injected into the targeted brain regions (dorsal striatum), and mice were put back in their housing regions after the injection. Researchers reported that pure GVs have high ultrasound contrast ability, theoretically, surface-modified GVs with intact shells should keep similar ultrasonic properties[71]. Therefore, we evaluated the stability of GVs and PGVs using ultrasound imaging. The ultrasonic contrast abilities of GVs and PGVs were then evaluated utilizing a Vevo 2100 imaging system (FUJIFILM VisualSonics Vevo LAZR Multi-modality) operating at 18 MHz, at various time intervals (0, 3, 6, 9, and 12 days) (Fig. 2b, Supplementary Fig. 1). The contrast mode image captures the ultrasound echo signals, and the quantitative intensity map is shown in Fig. 2c.

To further confirm the lifetime and stability of our GVs and PGVs in vivo, we also performed NIR fluorescent imaging with ICG-labeled GVs/PGVs- (ICG-GVs/ICG-PGVs) injected mouse brain by using the in vivo imaging system (IVIS) (Excitation/Emission wavelength: 780/805 nm, The Perkin-Elmer IVIS Lumina Series III)[57]. The survival time of GVs/PGVs in the mice brains was observed at 0, 3, 6, 9, 12, and 15 days using the IVIS (Fig. 2d, Supplementary Fig. 2d).

To evaluate the overall safety of the PGVs, we monitored the body weight of the PGVs-injected mice daily (Fig. 7a). Monitoring the change

in mice's weight acts as an overall pointer to health. We also monitored the general activities of mice to evaluate the brain function after the saline or PGVs administration.

## Open field test

An open field test (OFT) was performed to evaluate locomotor activity when mice were exposed to the novel environment. Standard protocol was conducted in a dark sound-attenuating apparatus to avoid outside interference. The activity was limited in the test chamber (40 cm length × 40 cm width × 30 cm height) with a white smooth floor[72]. Mice from the control group 5–7 days after saline injection into the striatum area and experiment group 5–7 days after PGVs-injection into the same brain region were gently placed in the center of the chamber and allowed to move freely for 5 min. The videos were recorded by a commercial digital camera (Logitech). The total distance traveled, and average speed were analyzed using an automated video tracking software ANY-maze (ANYmaze Video Tracking System 7.20).

## Y-maze

The Y-maze apparatus was made of gray polyvinyl chloride with 3 symmetrical arms ($30 \times 10 \times 15\,cm^3$) without extra- or intra-maze spatial cues and was evenly illuminated[73]. During the first trial (adaptive phase; 10 min), each mouse was allowed to explore 2 of the 3 arms with the third arm blocked (Supplementary Fig. 10a). After a one-hour inter-trial interval, each mouse was placed in the start arm of the Y-maze which is the same as the adaptive phase, and allowed to explore all arms freely (test phase; 5 min). An arm entry was counted when the main body of the mouse entered an arm. The percentage of time spent in the novel arm and the 2 familiar arms was calculated, with a higher preference for the novel arm being defined as intact spatial recognition memory.

## Novel object recognition

The experimental setup utilized for this study consisted of an open field plastic container with dimensions of 40 × 40 × 30 cm. Objects of comparable dimensions but varying shapes and colors were employed in the experimental procedures (Supplementary Fig. 10b, c). During the habituation phase, each mouse underwent a 10-min exposure within the enclosure and was subsequently returned to its home cage. On the subsequent day, two objects of similar nature were introduced, permitting the mice a 5-min interaction period before being returned to their home cages[74,75]. Following a 24-h resting interval, one of the objects was replaced with a novel counterpart, and the mice were reintroduced into the enclosure to engage with the objects for 5 min (Supplementary Fig. 10b). To eliminate the olfactory traces of the previous mice, both the objects and the container were meticulously cleansed with 70% ethanol after each trial. The spatial arrangement of the objects remained consistent between the training and testing phases. The calculation of the percentage of time spent on the novel object relative to the total time spent on both objects was expressed as $T_{Novel}/(T_{Novel} + T_{Old}) \times 100$, where $T_{Novel}$ represents the time allocated to the novel object, and $T_{Old}$ represents the time devoted to the familiar object. The data were analyzed by ANY-maze software.

## Mouse limb movement test

After delivering the PGVs or saline, mice were anesthetized with 2% isoflurane and eye ointment applied to both eyes. The mouse head was shaved, and the ultrasound transducer was attached to the skull and interfaced with ultrasound gel. Mice were stimulated with 0.17–0.54 MPa ultrasound (1.0 MHz) with 5 s intervals. Kinematic features of the forelimb during reaching behaviors were assessed with a commercial camera and the modeling tool TRACKER. In this system, video recordings were made with the commercial camera placed in front of the anesthetic mice, to trace and analyze forelimb movements during the ultrasound stimulation. Recordings were acquired by the

TRACKER software (TRACKER 6.0.2), and the movements of the paw were traced automatically to determine their coordinates. A 'zero' location was defined as the original position of the mouse limb, and '1' was defined as the base of the bench surface. The limbs' downward movement distance during US stimulation was then calculated as a percentage of the total movement between these two points. It was calculated as the y-axis percentage. Before each session, both the X-Y axis, scale bar, and the initial coordinate of the paw were manually labeled. Then the software can automatically trace the movement of the forelimb and calculate the coordinates. In another control experiment, collapsed PGVs were delivered to the right cortex area, and the same ultrasound conditions were used to sonicate the entire brain. 8 or 9 mice were used in the test for different treatment groups, and they each underwent 5–10 sonications in succession. A similar treatment protocol was used to record and examine the mouse fore-limb movements. After the treatment, the mice were put back in their original housing. The data were single-blinded, with the person evaluating the data being blinded to the treatment groups.

## EMG recording in anesthetized mice

PGVs/Saline/collapsed PGVs-injected mice were anesthetized with 2% isoflurane and eye ointment applied to the eyes. To track the bioelectric potential variations throughout mouse muscle tissue, two EMG electrodes were placed into the left forelimb's tricep muscle at a distance of 3–5 mm. The mouse's tail was equipped with an EMG ground wire. The isoflurane concentration was lowered to 0.5% before the EMG recording began (Medusa 1.06.01). Each individual received 3 rounds of ultrasound stimulation (1.0 MHz, 0.05–0.17 MPa). Each round had 7–10 ultrasonic stimuli that were spaced 3 s apart. There was a 50-s break between each stimulation round for each mouse. Using a multi-channel signal acquisition apparatus, EMG signals were captured (Medusa, Bio-Signal Technologies).

MATLAB was used to analyze EMG data (The MathWorks, Inc.). Using a 50 Hz notch and a 10–150 Hz bandpass filter, raw EMG data were filtered. The recording's beginning and a 200 ms window were chosen as the "calm time", and the signals in the stimulated period were corrected to these mean-filtered signals. The upper root-mean-square envelope with a 200 ms sliding window was used to smooth the data. The signal baseline for the recording was taken to be the mean of the corrected data during the calm period. The threshold was set at 1.2 times baseline, and any signal soaring over such level for at least 40 ms was regarded as a "spike". The ratio of the peak value of the recognized EMG event to the signal baseline, divided by the same baseline, is used to compute relative amplitude ($\Delta A/A_{base}$). The time interval between the ultrasonic stimulus being delivered and the moment the EMG signal crossed the threshold was used to calculate the reaction latency. A two-tailed unpaired t-test with Holm-Sidak correction was used to assess the statistical significance of these data.

## Fiber photometry recording

Anesthetized mice were fixed in a stereotaxic apparatus. A small section of the skull was removed (0.5 mm × 0.5 mm) after removing the hair and skin. An optical fiber was then implanted into the dorsal striatum or DRN. The fiber was fixed to the skull with dental cement and allowed to be set for 20 min. Mice were then moved back to their original housing for up to 5 days.

After the implantation and subsequent recovery period, mice were anesthetized with isoflurane. Ultrasound gel was administered to the depilated region of the cranium to enhance acoustic coupling. Mice were stimulated with 3 trials of ultrasound stimulation and each trial was 5–10 ultrasound stimuli, with 3 s or 5 s intervals between each pulse. Mice were allowed to rest for 50 s between trials. jRGECO1a/5-HT1.0 sensor fluorescence was captured with a fiber photometry system (Thinker Tech Nanjing BioScience Inc). The excitation wavelength for fiber photometry was 580 nm (jRGECO1a fluorescence) or 470 nm

(5-HT1.0 sensor fluorescence). Data were collected at 100 Hz and analyzed using a customized MATLAB script.

### In vivo ultrasound stimulation and mice behavior recording

Mice were injected with 1.0 μL PGVs in the right striatum (PGVs⁺ mice). Mice for the control group were injected with an equal volume of saline into the same brain region. A small wearable ultrasound transducer was customized with a frequency of 1.0 MHz. The transducer was packaged with a 3D-printed cap, and the cap was devised to suit the mounting plate which was stuck on the mouse head (Fig. 4a). This mounting plate was located above a targeted brain area. Before the behavior examination, the plate was loaded with degassed ultrasound gel, and the capped ultrasound transducer was inserted into the mounting plate. After habituation to the behavioral test environment for a few hours, the mice were evaluated for their behavioral responses to ultrasound stimulation. In each experiment, mice were put in a circular viewing space.

A camera (Logitech) was employed to track the motion behavior of mice before, during, and after ultrasound stimulation. In a behavioral assessment by software, the location of the white mounting plate was tracked automatically using TRACKER. For the rotation motor response, every second frame was analyzed and obtained 25 position measurements every second. The coordinates are generated from software to show the spatial motion XY-plots of the movement. The averaged linear and angular speed, and angular displacement were analyzed and matched between PGVs⁺ and Saline⁺ mice. The start of rotation was identified as the angular velocity > (mean + 3 × SD) within the five-second window before the ultrasound was turned on. Rotation latency was analyzed as the time delay from the onset of ultrasound to the onset of mouse rotation. For motion tracking of a freezing activity, the mouse head position was traced 25 times per second, and then the mouse's velocity was analyzed as the value of the linear velocity averaged within 3 s.

### Immunohistochemical staining

After delivering the PGVs or Saline, a 1.0 MHz transducer was coupled to the head with ultrasound gel. Mice were treated with ultrasound for 30 min under an anesthetized state. Anesthetized mice were transcardially perfused with PBS and 4% paraformaldehyde (PFA) (cat. no. P1110, Solarbio) 90 minutes after the conclusion of ultrasound treatment. Isolated brains were immersed in 4% PFA for an overnight fixation, and coronal slices measuring 40 μm in thickness were sectioned using a vibratome. Specific brain sections were gathered and subjected to a 90-minute blocking process in a blocking medium (0.3% TritonX-100 and 10% normal goat serum with 1% BSA) followed by overnight incubation with primary antibody at 4 °C. Following three 10-minute PBS washes, the brain slices underwent a 2-hour incubation with the secondary antibody at room temperature. After three additional PBS washes, the slices were mounted on glass slides using droplets of Prolong Diamond Antifade Mountant with DAPI. Primary antibodies used were MAP2 (PA1-10005, Invitrogen, diluted 1:1,000), Caspase-3 (#9661, Cell Signaling Technology, diluted 1:400), iba-1 (#17198, Cell Signaling Technology, diluted 1:500), GFAP (#12389, Cell Signaling Technology, diluted 1:400), TPH2 (#51124, Cell Signaling Technology, diluted 1:1000) and c-Fos (#2250, Cell Signaling Technology, diluted 1:500). Secondary antibodies used at a dilution of 1:1,000 were goat anti-chicken IgY (H + L) Alexa Fluor 555 (A-21103, Invitrogen) or goat-anti-rabbit IgG (H + L) Alexa Fluor 488 (A-21428, Invitrogen). The number of c-Fos⁺ nuclei, and Caspase-3, iba-1, and GFAP cells were counted using ImageJ, and the number of c-Fos⁺ cells per 200 × 200 μm area of slice; the 3 other proteins were counted using slices of 450 × 450 μm. The counting of c-Fos⁺, Caspase-3⁺, iba-1⁺, and GFAP⁺ cells was single-blinded and performed by an experimenter who did not know the groups beforehand. All brain slices were imaged using a confocal microscope (TCS SP8, Leica) or live-cell fluorescence microscope (Nikon Eclipse Ti2-E) in the ULS facilities in The Hong Kong Polytechnic University.

### Modeling depression in mice

C57BL/6 mice were individually put into a customized well-ventilated 50 mL tube to give restraint. Mice could not move forward or backward in our tailored tubes. This restraint stress was given to mice for 3 h daily (set times: 1 PM - 4 PM). The Control group mice stayed in normal conditions without any restraint stress. After the 3-h restraint stress management, confined mice were returned to their regular house conditions. This condition was maintained for 2 weeks (Fig. 6a).

### Behavior tests by tail suspension and forced swimming

For the depression-related behavior tests (tail suspension test (TST)), and forced swimming test (FST)), mice were exposed to unavoidable stress by hanging their tails, placing them in water and finally calculating the immobile period. For TST, the distal end of the tail was attached to the ceiling of a test box (40 × 40 × 40 cm). Mice were suspended for 6 min (40 cm from the floor), including 2 min of habitation and 4 min of testing, and the overall immobile period was analyzed by another experimenter who was blinded to the treatment groups. During the experiments, the behavior of mice was videotaped. Each mouse's performance was manually assessed after the tests, and mice who supported their hind limbs with forelimbs or crawled on their tails were ignored during the evaluation[76–78].

FST was performed 1–2 days after TST. Mice were put separately in a transparent cylinder in water (diameter: 20 cm, depth: 20 cm, room temperature) in which they could not touch the bottom of the beakers for 6 min. A mouse was considered stationary when it floated on the water in a vertical position and made only minimal movements to maintain its head beyond the surface of the water. The experimental course was recorded by a camera in front of the chamber, and the resting state was analyzed during a final 4 min in a 6 min test time by a single-blinded experimenter. Mobility was identified as horizontal movement and upright movement of the forelimbs. After testing, mice were placed in a pre-warmed cage for 30 min after which they were returned to their home cages.

### Statistical analysis

Statistical analyses and graph preparation were conducted using the GraphPad Prism software 8.0.1. Two-tailed unpaired t-tests, as well as one- or two-way analysis of variance (ANOVA), were employed to ascertain statistical significance, with the application of post-hoc tests or corrections where deemed suitable. Statistical significance was considered at $P$ values below 0.05. Specific details regarding the applied statistical tests for each panel are elucidated in the figure legends. The presentation of data is expressed either as mean ± standard error of the mean (S.E.M.) or mean ± standard deviation (S.D.), as specified.

### Reporting summary

Further information on research design is available in the Nature Portfolio Reporting Summary linked to this article.

## Data availability

The main data supporting the results of this study are available within the paper and its Supplementary Information. The raw and analyzed datasets generated during the study are too large to be publicly shared, yet they are available for research purposes from the corresponding author upon reasonable request. Source data are provided in this paper. Source data are provided with this paper.

## Code availability

Custom code is provided in the Supplementary information.

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

## Acknowledgements

This work was supported by the Hong Kong Research Grants Council Collaborative Research Fund (C5053-22GF), General Research Fund (15224323 and 15104520), Hong Kong Innovation Technology Fund (MHP/014/19), Shenzhen-Hong Kong-Macau Science and Technology Program (SGDX20201103095400001), internal funding from the Hong Kong Polytechnic University Research Institute of Smart Ageing (1-CD76) and Hong Kong Polytechnic University (1-ZVW8). The authors would like to thank the facility and technical support from the University Research Facility in Life Sciences (ULS) and University Research Facility in Behavioral and Systems Neuroscience (UBSN) of The Hong Kong Polytechnic University. Figures 2a, 3a, g, 5a, 6a, and Supplementary Figs. 2a, 3a, 9a, 10a, b were created with BioRender.com.

## Author contributions

XD.H., and J.J. contributed equally to this work. XD.H. and L.S. contributed to the conceptualization; XD.H., J.J., T.L., Q.X., J.Z. and L.S. contributed to the methodology; XD.H., J.J., K.F.W., Y.J., M.S., X.Z., XH.H. and D.L. contributed to the investigation; XD.H., J.J., K.F.W., L.L. and Y.J. contributed to the data analysis; XD.H., J.J., Z.Q. and L.S. contributed to the manuscript preparation; L.S. contributed to the funding acquisition.

## Competing interests

The authors declare no competing interests.
