## [Peer Review File · Nature Communications]

Nanobubble-actuated ultrasound neuromodulation for selectively shaping behavior in miceReviewers' comments:

Reviewer #1 (Remarks to the Author):

This work by Hou et al. presents a significant improvement in the development of ultrasound-based neuromodulation technologies. Conventional ultrasound neuromodulation and "sonogenetics" approaches are limited by their inadequate spatial resolution and requirement of high peak pressures to achieve physiological effects. To mitigate these challenges, in this paper the authors demonstrate a novel approach to "sensitize" the targeted region to incident ultrasound by delivering nanobubbles. Specifically, the authors used PEGylated gas vesicles to enhance the local response of endogenous mechanosensitive ion channels to low-pressure ultrasound. To prove the efficacy of this approach, the authors used a large array of experiments, ranging from calcium imaging, c-fos immunostaining, limb movement, electromyography, inducing locomotor and freezing behaviors, to relieving depression symptoms in mice. The authors also demonstrated the lack of auditory pathway activation as an alternative mechanism contributing to the observed effects. Overall this work represents an important step in the development of ultrasound neuromodulation technologies, and the paper is comprehensive and well-written. I recommend acceptance of this work after a few comments below are addressed:

What is the long-term utility of the injected gas vesicles? After injection, what is the maximum duration for which these gas vesicles remain effective in eliciting neural responses under FUS? It may be helpful to plot the minimum required peak pressure of FUS as a function of days after gas vesicle injection to find out when injected gas vesicles become ineffective.

There are many endogenous mechanosensitive ion channels that may respond to FUS. What ones contribute to neural activation in the presence of these gas vesicles, and what ones contribute to neural activation in the absence of these gas vesicles?

The authors proposed the application of this approach in non-human primates. However, invasive injections of these gas bubbles are still required for a chronic study, which may incur repeated damage to the brain tissue. Also, how much will the incident ultrasound become attenuated after passing through the skull of NHPs?

Reviewer #2 (Remarks to the Author):

This manuscript by Hou et al. reported the combination of ultrasound with nanobubbles to achieve localized neuromodulation that can reach the deep brain. This work is built on the team's earlier published work in *Advanced Science* (<https://doi.org/10.1002/adv.202101934>). They showed that increased c-Fos expression was observed in the sub-millimeter-scale region, demonstrating the potential of the proposed technique to achieve high spatial resolution. This technique was used to selectively activate 5-HT neurons in the dorsal raphe nucleus to reduce depression-like behaviors in a mouse model. Although extensive experiments were performed in vitro and in vivo, the reviewer thinks that this work does not represent a sufficiently striking advance in the field of ultrasound neuromodulation. More details of the major weaknesses are provided below.

1. The difference between gas vesicles (GVs), which were called nanobubbles in the title, used in this manuscript and those used in their previous *Advanced Science* paper is that the GV's were coated with PEG, called PGVs, to improve biocompatibility. The PET coating is not expected to change the acoustic property of the GV's significantly. However, the representative frequency spectrum of the backscattered signals from PGVs is clearly different from that shown in the *Advanced Science* paper for GV's. GV's are nanometers in size with a resonance frequency that was reported to be extremely high (e.g., ~1 GHz). At 1 MHz, these GV's are not strong scatterers of ultrasound and don't expect to have strong nonlinear oscillations, especially at the low driving acoustic pressures used in this study. However, Fig. 1e shows not only strong harmonics but also strong subharmonics. Only one representative frequency spectrum is presented without data from the control condition and repeated measurements. This left the reviewer wondering whether these

GVs and PGVs are truly responsive to ultrasound at 1 MHz with such low acoustic pressures. It is critical to present the frequency spectrum of the acoustic emissions detected in vivo in mice at a PGV injection concentration that is safe for neuromodulation to provide convincing data to support that these PGVs can amplify ultrasound sonication at 1 MHz frequency and extremely low acoustic pressures.

2. One critical concern of using PGVs for neuromodulation is safety. It was stated in this manuscript that the mice had steady body weights, and none died. This is not sufficient to justify the safety of the PGVs. Supplementary Fig. 8 presented immunohistochemistry staining of Iba1, GFAP, and Caspase-3. Without confirming these brain regions that were quantified indeed had PGVs, these staining results are not strong evidence to support the safety of PGVs. The observation that PGV injection alone had some effects on motor behavior (Fig. 4c) further raises safety concerns. Safety evaluations for neuromodulation should not be limited to histological analysis. It is important to verify that agents injected into the brain do not change the normal function of the brain.

3. The observation that c-Fos expression was confined to the sub-millimeter region is interesting. C-Fos expression was spread to a millimeter-scale region in their Advanced Science paper, but it was confined to a sub-millimeter region in this paper. It is important to show the co-localization of the injected PGVs with the observed c-FOS signals to confirm that the increased c-FOS expression was indeed associated with the injected PGVs. No discussion is provided for why the region could reach the sub-millimeter scale.

4. The low animal number in each group (n=3) limits the rigor of the study design, especially considering each mouse was used to test multiple acoustic pressures in some of the studies.

5. It is mentioned that PGVs+US evoked greater EMG responses than their previous findings using MscL-sonogenetics, and a similar phenomenon was found when comparing PGVs+US vs. TRPA1-sonogenetics. TRPA1-sonogenetics study performed limb movement tests and EGM recording but did not perform the same quantifications as this study. It is not clear how this manuscript reached the conclusion that PGVs+US could potentially be better than TRPA1-sonogenetics. Furthermore, the conclusion that PGVs+US is more promising for clinical translation than sonogenetics is an overstatement. Genetically engineering carries risk but also offers the benefit of long-term expression.

6. It is claimed that this strategy is potentially well-suited for deep brain stimulation in non-human primates. If one injection of the PGVs only leads to a sub-millimeter region activation of the neurons, this technique would not be suitable for activating the large-scale non-human primate brain.

Reviewer #1:

This work by Hou et al. presents a significant improvement in the development of ultrasound-based neuromodulation technologies. Conventional ultrasound neuromodulation and “sonogenetics” approaches are limited by their inadequate spatial resolution and requirement of high peak pressures to achieve physiological effects. To mitigate these challenges, in this paper the authors demonstrate a novel approach to “sensitize” the targeted region to incident ultrasound by delivering nanobubbles. Specifically, the authors used PEGylated gas vesicles to enhance the local response of endogenous mechanosensitive ion channels to low-pressure ultrasound. To prove the efficacy of this approach, the authors used a large array of experiments, ranging from calcium imaging, c-fos immunostaining, limb movement, electromyography, inducing locomotor and freezing behaviors, to relieving depression symptoms in mice. The authors also demonstrated the lack of auditory pathway activation as an alternative mechanism contributing to the observed effects. Overall this work represents an important step in the development of ultrasound neuromodulation technologies, and the paper is comprehensive and well-written. I recommend acceptance of this work after a few comments below are addressed:

Reply: We appreciate the reviewer’s comments, and sincerely acknowledge the favorable recommendations.

1.) What is the long-term utility of the injected gas vesicles? After injection, what is the maximum duration for which these gas vesicles remain effective in eliciting neural responses under FUS? It may be helpful to plot the minimum required peak pressure of FUS as a function of days after gas vesicle injection to find out when injected gas vesicles become ineffective.

Reply: Long-term stability of PGVs can reduce the number of repetitive injections needed for chronic brain disease treatment, thereby decreasing brain tissue damage. In principle, this would mean that a single administration of PGVs could effectively mediate multiple rounds of US-based neuromodulatory treatment without the need for further surgical invasion. Consequently, this property can help researchers expand their studies of US as a

therapeutic mechanism and enable comparisons between, for instance, single and multiple rounds of treatment, while remaining confident of effective targeting of the desired site. In the present study, ultrasound imaging was detected up to 12 days post-injection and fluorescence signals of PGVs were detectable on day 15 (Manuscript Fig. 2b,d, below), indicating that PGVs are present and stable for at least 12 days *in vivo*. We would like to emphasize that the presence of the ultrasound signal itself serves as a confirmation of the PGVs' integrity *in situ* and their ability to serve as mediators for ultrasound neuromodulation. This is because broken or damaged PGVs lose their ability to produce contrast signals and, consequently, their neuromodulatory effect.

We agree with the reviewer that the PGVs' effectiveness needs to be evaluated through more behavioral experimentation and have conducted additional experiments to “find out when injected PGVs become ineffective” to enhance our study. We **quantified the PGVs+US-evoked neural calcium responses via fiber photometry *in vivo* and found that the corresponding calcium signals persisted until day 12, indicating that PGVs' effective time was up to 12 days** in Figure R1a,b (below). Such results are in line with our PGVs imaging data (manuscript, Figure 2b-d, below). We have added such results as a new Supplementary Figure 6 in a resubmitted manuscript as a response to the reviewer's valuable comment. You can find the new data **indicated in yellow highlighter**.

Figure R1. a) Mean jRGECO1a fluorescence (Ca^{2+} signals) traces from PGVs+ mice under US stimulation. The light-yellow box represents the US pulse. b) Mean peak of Ca^{2+} responses under 0.09 MPa US irradiation at several injected days. N = 9 mice.

Figure 2b-d (Manuscript). b) Ultrasound contrast-mode images of PGV-injected mice brains at several injected days. PGVs collapsed with destructive insonation. c) Quantitative results of the PGVs and GVs signals in mice brains by ultrasound imaging and IVIS. Data represent the mean \pm SEM from 3 independent experiments. d) NIR fluorescent imaging of mice brains with several injected-time points (0, 3, 6, 9, 12, and 15 days) with brain ICG-PGVs injection.

2.) There are many endogenous mechanosensitive ion channels that may respond to FUS. What ones contribute to neural activation in the presence of these gas vesicles, and what ones contribute to neural activation in the absence of these gas vesicles?

Reply: Neurons have been reported to express multiple channels with mechanosensitivity, including TRPV1, TRPV2, TRPV4, Piezo1, TRPC1, TRPM7, and the TRPP1/2 complex, as mediators for the ultrasound neuromodulation effect.¹ There is not yet any consensus about the degree of each channel's contribution to neuromodulatory effects, but it is believed that they are not equally important, given the significant variation in their respective expression levels and required activation thresholds. At the neuronal level, we

do not have reason to believe that PGV injection would change the endogenous levels and dynamic characteristics of these channels, nor the mechanisms through which neurons respond to ultrasound. However, PGVs do produce more harmonics than ultrasound alone which may activate some channels more easily. We have not encountered evidence or literature showing that mechanosensitive ion channels are frequency-dependent in the range of PGV harmonics. Therefore, we tend NOT to believe that PGVs may selectively impact some channels in preference to others. However, this possibility can certainly not be ruled out at this point, and needs more investigation.

In contrast, given the high sensitivity and expression of the Piezo1 channel, our group has shown that Piezo1 played a major role in response to ultrasound stimulation in our recent PNAS paper.²

3.) The authors proposed the application of this approach in non-human primates. However, invasive injections of these gas bubbles are still required for a chronic study, which may incur repeated damage to the brain tissue. Also, how much will the incident ultrasound become attenuated after passing through the skull of NHPs?

Reply: We agree that the application of this technology to non-human primates may be premature and difficult because of the invasive injection of these PGVs. Based on our research, we tend to believe that PGVs, with their greater longevity *in vivo*, would actually reduce precisely such brain damage as the reviewer speaks of. However, we do agree that this proposition remains to be tested in non-human primate models. Therefore, in order not to engage in evidence-free speculation, we have **removed statements to this effect from the Discussion section** in the accompanying manuscript.

Reviewer #2:

1.) The difference between gas vesicles (GVs), which were called nanobubbles in the title, used in this manuscript and those used in their previous Advanced Science paper is that the GV's were coated with PEG, called PGVs, to improve biocompatibility. The PEG coating is not expected to change the acoustic property of the GV's significantly. However, the representative frequency spectrum of the backscattered signals from PGVs is clearly different from that shown in the Advanced Science paper for GV's. GV's are nanometers in size with a resonance frequency that was reported to be extremely high (e.g., ~1 GHz). At 1 MHz, these GV's are not strong scatters of ultrasound and don't expect to have strong nonlinear oscillations, especially at the low driving acoustic pressures used in this study. However, Fig. 1e shows not only strong harmonics but also strong subharmonics. Only one representative frequency spectrum is presented without data from the control condition and repeated measurements. This left the reviewer wondering whether these GV's and PGVs are truly responsive to ultrasound at 1 MHz with such low acoustic pressures. It is critical to present the frequency spectrum of the acoustic emissions detected *in vivo* in mice at a PGV injection concentration that is safe for neuromodulation to provide convincing data to support that these PGVs can amplify ultrasound sonication at 1 MHz frequency and extremely low acoustic pressures.

Reply: We greatly appreciate the reviewer's efforts to examine our research comprehensively. PEG is widely used for surface modification, and we concur that it should not have a significant effect on the acoustic properties of GV's. We also agree that there are some differences in the acoustic frequency spectrum between PGVs and GV's in our results, which we believe could be caused by the lower concentration of PGVs used in this study in the tested solution (0.4 nM, half the amount of GV's in the Advanced Science paper).

Our previous study has shown that the resonant frequency of GV's is around 100 MHz (much lower than 1 GHz).³ In addition to our Advanced Science paper, which showed clear activation of *in vitro* neurons and *in vivo* brains, Bar-Zion's study also demonstrated that GV's can respond to **0.67 MHz** ultrasound (similar to 1 MHz).⁴ Moreover, GV's can act as ultrasound contrast agents with non-linear signals up to 18 MHz far from their resonance.⁵

In addition, the present study also shows the acoustic signals generated under 1 MHz ultrasound stimulation (Manuscript Fig. 1e). These results demonstrate that GVs are not necessarily driven in their resonant frequency to have a response. It is understood that the non-linear oscillation due to the buckling effects is the major reason for its acoustic behaviors, which has a wide frequency spectrum.⁶

However, the reviewer's point is well-taken. We **conducted an additional experiment to examine the acoustic signals from the mice brains after PGVs injection** (below Figure R2a,b), with the same amount (8.0 nM) as our manuscript *in vivo* experiments. Similar harmonic signals were detected under 1 MHz ultrasound, which was quantified by comparing them with the control group (saline-injected mice). The below results corroborate that PGVs can indeed respond to 1 MHz ultrasound (Figure R2c).

Figure R2. Passive cavitation detection in the mouse brain with/without PGVs injection. a) Averaged frequency spectrum of backscattered signals from a PGVs-injected mouse brain. b) Representative time-domain waveform of backscattered signals from PGVs- and saline-injected mouse brains sonicated by a 1.0 MHz ultrasound transducer. c) Quantified results of broadband signals, as in (b). Bars represent mean \pm SEM from three independent experiments. ** $p < 0.01$, two-tailed unpaired t-test.

2.) One critical concern of using PGVs for neuromodulation is safety. It was stated in this manuscript that the mice had steady body weights, and none died. This is not sufficient to justify the safety of the PGVs. Supplementary Fig. 8 presented immunohistochemistry staining of Iba1, GFAP, and Caspase-3. Without confirming these brain regions that were quantified indeed had PGVs, these staining results are not strong evidence to support the safety of PGVs. The observation that PGV injection alone had some effects on motor behavior (Fig. 4c) further raises safety concerns. Safety evaluations for neuromodulation should not be limited to histological analysis. It is important to verify that agents injected into the brain do not change the normal function of the brain.

Reply: We appreciate this constructive critique of our data. To demonstrate the selected

brain regions indeed had PGVs, we provide an enlarged version of manuscript Fig. 7 (below) to address the reviewer's concern. Note that the cell nuclei were intact after Saline/PGVs+US stimulation (Figure R3, below). No significant difference in the expression of Iba1, GFAP, and Caspase-3 (red) was found between the +US and -US groups (Figure R3, below). PGVs (green) injection also did not cause any significant changes in the percentages of microglia, astrocytes, and apoptotic cells. Based on these results we also did not find obvious adverse effects of PGVs.

Figure R3. Evaluation of neural inflammation, and apoptosis after US exposure in the US-treated brain region using immunohistochemical staining of microglia (Iba1), astrocytes (GFAP), and Caspase-3, green shows the injected PGVs, blue indicates the DAPI-stained nuclei.

To better identify the synchronized signals of PGV+US-treated mice motor changes, we smoothed the mice motor behaviors' data, however, such behaviors' change results seem to be over-smoothed. We believe that the apparent differences in motor behavior observed by the reviewer were likely due to the method we used to perform the smoothing of data (Manuscript Fig. 4c). We thank the reviewer for pointing this out, and **providing a**

new, more representative picture with appropriate data smoothing in the revised manuscript to make this point (Figure R4, below). We hope these improved figures are sufficient to quell the reviewer's doubts.

Figure R4. (Top) A plot of the angular displacement showing a mouse turning during PGVs+US stimulation. Counter-clockwise angular changes were counted as positive changes in angles. During US onsets, the mouse turned unilaterally significantly more than pre-US. (Bottom) A trace shows the angular speeds of the mouse. Speed markedly increases during US onsets (dotted lines).

For the safety of this new strategy, we used well-established methods to examine the mice's overall health and molecular markers thereof and did not find obvious adverse effects. Certainly, these data cannot rule out the possibility of some specific safety concerns. However, for this new strategy, given the current data, we do not believe that PGVs pose a significant safety issue. Here, we are happy to provide general activity tests to verify the mice's brain function, such as **locomotor activity (Figure R5a,b, below), memory (Figure R5c,d, below), or cognition (Figure R5e,f, below) to further bolster this point.** **We monitored the mice's behaviors after PGVs treatment and found that PGVs did not affect normal brain function compared with the saline-injected mice group.** We have added a new Figure 7 about the PGVs safety test in a resubmitted manuscript as a response to the reviewer's valued comment. You can find the new data **indicated in yellow highlighter.**

Figure R5. a) Representative trajectories recorded from mice injected with Saline or PGVs into dorsal striatum (each trace 5 min long). No side effects were observed after PGVs administration. b) Total distance traveled in open field test (OFT) 5 days after injection of PGVs or control mice. $n=6$ mice in each group. Two-tailed unpaired t-test. c) Heatmap of the mice movement in Y maze. d) Calculated percentage of time spent by mice on the three arms (start, other, novel areas) in Saline⁺ and PGVs⁺ mice. $n = 6$ mice in each group. Two-tailed unpaired t-test with Holm–Sidak correction. e) Representative trajectories recorded from Saline⁺ mice (left) and PGVs⁺ mice (right) during the novel object recognition (NOR) test. f) Number percentage of object investigation by saline⁺ or PGVs⁺ mice. n.s., not significant, Data are shown as mean \pm SD.

3.) The observation that c-Fos expression was confined to the sub-millimeter region is interesting. C-Fos expression was spread to a millimeter-scale region in their Advanced Science paper, but it was confined to a sub-millimeter region in this paper. It is important to show the co-localization of the injected PGVs with the observed c-FOS signals to confirm that the increased c-FOS expression was indeed associated with the injected PGVs. No discussion is provided for why the region could reach the sub-millimeter scale.

Reply: The animal experimental procedures in these two studies have important differences between them.⁷ Aside from the differences between GVs (Advanced Science paper) and PEGylated GVs, in the present manuscript, mice were allowed a 5-day recovery period after PGV injection, whereas c-Fos data in the Advanced Science paper were from mice treated shortly after GVs were delivered into the brain. Given the disparity in post-injection times, some PGVs may have been degraded and cleared from peripheral areas in the intervening period, making the impacted area smaller and more localized. **Regarding the co-localization of PGVs and c-Fos, we already report the co-localization of c-Fos and PGVs in the present manuscript, Figure 3k. We have added a relevant discussion about the spatial resolution in our Discussion section as was mentioned in the comment.**

You can find the changes **indicated in yellow highlighter in a resubmitted manuscript.**

Figure 3k (Manuscript). c-Fos expression (red) around PGVs injection area (green) in mouse cortex stimulated with PGVs+US.

4.) The low animal number in each group (n=3) limits the rigor of the study design, especially considering each mouse was used to test multiple acoustic pressures in some of the studies.

Reply: We appreciate and concur with the reviewer’s comment. To strengthen our study, we have performed our experiments with more animals, and **most of the behavioral tests using mice number n = 9/10, the minimum n = 6.** You can find mice number **N indicated in yellow highlighter in a resubmitted manuscript.**

5.) It is mentioned that PGVs+US evoked greater EMG responses than their previous findings using MscL-sonogenetics, and a similar phenomenon was found when comparing PGVs+US vs. TRPA1-sonogenetics. TRPA1-sonogenetics study performed limb movement tests and EGM recording but did not perform the same quantifications as this study. It is not clear how this manuscript reached the conclusion that PGVs+US could potentially be better than TRPA1-sonogenetics. Furthermore, the conclusion that PGVs+US is more promising for clinical translation than sonogenetics is an overstatement. Genetically engineering carries risk but also offers the benefit of long-term expression.

Reply: For the comparison to TRPA1-sonogenetics, we would like to first point out that our studies required a lower ultrasound intensity than the TRPA1-study. We realize that such a statement was more expressive of our subjective, though experimentally-informed,

opinion than an established fact. We may have expressed our point a tad inartfully, but the intention was to stress that the approach we describe does not compete with genetically-based sonogenetic strategies, and could possibly even be used in conjunction with them. We simply hope that this could provide a different path to achieving sonogenetics *in vivo*. We apologize for any confusion this may have caused in our manuscript. To avoid creating any further confusion or misstatement, we have **removed the portion concerning the comparison to TRPA1-sonogenetics** in the accompanying revised manuscript.

6.) It is claimed that this strategy is potentially well-suited for deep brain stimulation in non-human primates. If one injection of the PGVs only leads to a sub-millimeter region activation of the neurons, this technique would not be suitable for activating the large-scale non-human primate brain.

Reply: The results shown in the present manuscripts demonstrate that ultrasound can control neuronal activities in the presence of PGVs, but not in the ultrasound-alone group. Therefore, the neuromodulation achieved is reliant upon PGVs, and the spatial precision of neuronal activation will be highly dependent on the injection site and survival region of PGVs. By the same principle as in small rodents, it should be possible to activate a specific brain region in other small animals or non-human primates: by injecting well-controlled amounts of PGVs to a targeted site as a way to preferentially sensitize it to ultrasound stimulation. However, we acknowledge our current study is still away from the non-human primate testing phase, and we have **removed the discussion of non-human primates to avoid confusion** in the revised manuscript.

Reference:

1. Yoo S, Mittelstein DR, Hurt RC, Lacroix J, Shapiro MG. Focused ultrasound excites cortical neurons via mechanosensitive calcium accumulation and ion channel amplification. *Nature communications* **13**, 493 (2022).
2. Zhu J, *et al.* The mechanosensitive ion channel Piezo1 contributes to ultrasound neuromodulation. *Proceedings of the National Academy of Sciences* **120**, e2300291120 (2023).
3. Yang Y, Qiu Z, Hou X, Sun L. Ultrasonic characteristics and cellular properties of Anabaena gas vesicles. *Ultrasound in medicine & biology* **43**, 2862-2870 (2017).

4. Bar-Zion A, *et al.* Acoustically triggered mechanotherapy using genetically encoded gas vesicles. *Nature nanotechnology* **16**, 1403-1412 (2021).
5. Wang G, *et al.* Surface-modified GVs as nanosized contrast agents for molecular ultrasound imaging of tumor. *Biomaterials* **236**, 119803 (2020).
6. Salahshoor H, *et al.* Geometric effects in gas vesicle buckling under ultrasound. *Biophysical Journal* **121**, 4221-4228 (2022).
7. Hou X, *et al.* Precise ultrasound neuromodulation in a deep brain region using nano gas vesicles as actuators. *Advanced Science* **8**, 2101934 (2021).

REVIEWERS' COMMENTS

Reviewer #1 (Remarks to the Author):

The authors have addressed all comments with satisfaction, and I hereby support its acceptance for publication.

Reviewer #2 (Remarks to the Author):

The authors have addressed my comments in the revised manuscript.